# PLASMA : PROCEDURAL KNOWLEDGE MODELS FOR LANGUAGE-BASED PLANNING AND RE-PLANNING

Faeze Brahman [12]   Chandra Bhagavatula [1]
Valentina Pyatkin [1†]   Jena D. Hwang [1†]   Xiang Lorraine Li [15]   Hirona J. Arai [3]
Soumya Sanyal [3]   Keisuke Sakaguchi [4]   Xiang Ren [13]   Yejin Choi [12]
[1]Allen Institute for Artificial Intelligence   [2]University of Washington
[3]University of Southern California   [4]Tohoku University   [5]University of Pittsburg
faezeb@allenai.org

## ABSTRACT

Procedural planning, which entails decomposing a high-level goal into a sequence of temporally ordered steps, is an important yet intricate task for machines. It involves integrating common-sense knowledge to reason about complex and often contextualized situations, e.g. "scheduling a doctor's appointment without a phone". While current approaches show encouraging results using large language models (LLMs), they are hindered by drawbacks such as costly API calls and reproducibility issues. In this paper, we advocate planning using smaller language models. We present PLASMA, a novel two-pronged approach to endow small language models with procedural knowledge and (constrained) language planning capabilities. More concretely, we develop *symbolic procedural knowledge distillation* to enhance the commonsense knowledge in small language models and an *inference-time algorithm* to facilitate more structured and accurate reasoning. In addition, we introduce a new related task, *Replanning*, that requires a revision of a plan to cope with a constrained situation. In both the planning and replanning settings, we show that orders-of-magnitude smaller models (770M-11B parameters) can compete and often surpass their larger teacher models' capabilities. Finally, we showcase successful application of PLASMA in an embodied environment, VirtualHome.[1]

## 1 INTRODUCTION

Powered by massive scale, large language models (LLMs) excel on many downstream tasks that require commonsense. One such task is *procedural planning* (Schank & Abelson, 1975b; Pearson & Laird, 2005), a task that involves decomposing a high-level **goal** into a sequence of coherent, logical, and goal-oriented steps (**plan**) (e.g. "see a movie" → "Look up movie showings", "Choose a movie" . . .). Recent approaches model this task as a conditional language generation problem using LLMs (Madaan et al., 2022; Huang et al., 2022; Ahn et al., 2022; Zhao et al., 2023). Despite their reasonable performance on the task, their steep computational cost and inaccessibility to models' parameters hinder the wider adoption of LLMs (OpenAI, 2023) for procedural planning.

We present PLASMA (PLAn with SMAll models), a novel framework and model to impart procedural knowledge and language-based planning abilities in small LMs.[2] In the *first phase* of the framework, we enhance the implicit commonsense knowledge in small LMs through **symbolic *procedural knowledge distillation*** (West et al., 2022; Bhagavatula et al., 2023) as illustrated in Figure 1. We formulate it in two stages: (i) *Knowledge verbalization* to generate procedural knowledge from an LLM, and (ii) *Knowledge distillation* to transfer LLM-generated knowledge to a smaller LM.

For the knowledge distillation stage, we introduce two constrained settings: *Constrained planning* and *Counterfactual replanning* in addition to the standard language planning task. These tasks enable

---

[†]Authors contributed equally.

[1]Our data and code is publicly available at: https://github.com/allenai/PlaSma

[2]Hereafter, we will use **'planning'** to refer to **'language-based planning'** for brevity.

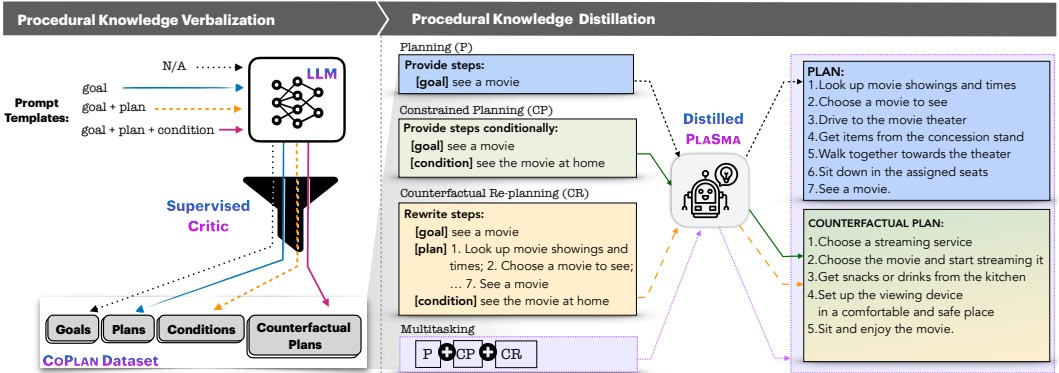

Figure 1: Symbolic Procedural Knowledge Distillation.

a more realistic setting by requiring models to reason about contextually constrained situations in real-world applications. Specifically, the model generates or revises a plan based on a given goal (e.g., "see a movie") while adhering to an additional **condition** (e.g., "at home"). Our knowledge verbalization process results in a large dataset for (i) language-based planning, (ii) language-based planning under constraints, and (iii) language-based re-planning of existing plans under constraints. Our dataset, CoPLAN, is then used to train smaller models, PLASMA, using both task-specific and multi-task distillation.

For the *second phase* of PLASMA, we enable structured, tree-based reasoning via a novel **inference-time decoding algorithm** (Figure 2). We observe that the standard next-token prediction objective in auto-regressive LMs (applied during distillation) does not equip them with sufficient causal and temporal reasoning abilities to generate high-quality plans, or a mechanism to rectify their mistakes in earlier steps. To address this challenge, we develop a *verifier-guided step-wise beam search* to better leverage the multi-step structure of plans (resulting in PLASMA+). Concretely, we incorporate a step-wise verifier in a tree-based decoding algorithm to guide PLASMA+ to generate more semantically coherent and temporally accurate plans.

Experimental results show that our approach is effective at endowing smaller LMs with planning abilities. For the standard planning task, smaller student models (of varying sizes) achieve 17.57% relative improvements, on average, over their teacher. The best student model is comparable even to GPT-3, a model 16 times the student's size. For the first time, we distill constrained and counterfactual planning abilities in small-size models, achieving 93% and 86% validity rates according to human evaluation. Interestingly, in the VirtualHome environment (Puig et al., 2018), our model significantly outperforms previous work based on GPT-3 (Huang et al., 2022) on executability (absolute 17%) and correctness (absolute 25%). Our framework including symbolic procedural distillation, decoding-time algorithm, and the proposed tasks and the accompanying CoPLAN dataset provide valuable resource and direction for advancing research in the field of procedural language-based planning.

## 2 SMALL LANGUAGE MODELS AS PROCEDURAL KNOWLEDGE MODELS

In this section, we discuss how to endow small student models with procedural knowledge for (constrained and counterfactual) planning capabilities. We first describe our knowledge verbalization and distillation framework which we collectively refer to as Symbolic Procedural Knowledge Distillation (§2.1, §2.2). We then propose a strategy to enhance the reasoning capabilities of small students via a novel verifier-guided step-wise decoding algorithm (§2.3).

### 2.1 CoPLAN: PROCEDURAL KNOWLEDGE VERBALIZATION FROM LARGE TEACHERS

Large language model can perform new tasks by adapting to a few in-context examples (Brown et al., 2020). We thus leverage this emergent reasoning capabilities of LLM to circumvent the challenge of crowdsourcing supervised datasets at scale. We collect data targeting the following three tasks:

1. **Goal-based Planning (pl.)**, decomposing a high-level goal $g$ into a sequence of temporally extended steps $y = \{s_t\}_{t=1}^T$.
2. **Constrained Planning (cp.)**, decomposing a high-level goal $g$ into a sequence of temporally extended steps $y = \{s_t\}_{t=1}^T$ while satisfying a given condition $c$.
3. **Counterfactual Replanning (cr.)**, rewriting an initial plan $y$ to a given goal $g$ into a new plan $y'$ in order to satisfy a given condition $c$.

Our knowledge verbalization pipeline shown in the left side of Figure 1 is a two-stage process: 1) instance generation through few-shot prompting, and 2) automatic data curation using a critic to filter out the low quality data. The process results in COPLAN, a quality dataset containing goals, plans, conditions, and counterfactual plans.

**Step 1. Data Generation** We start by generating a large pool of goals $\mathcal{G}$ with a diverse range of topics in a bootstrapping fashion. Concretely, we start with 5 manually written goals and expand them through prompting GPT-3. We then manually filter out low-quality (in terms of acceptability/achievability) ones and repeat this expansion/filtering for several iterations until we obtain a seed goal pool with 100 goals. We subsequently use this goal pool for randomly selecting few-shot examples for prompting and generating a large number of goals in our final dataset.

For each generated goal $g \in \mathcal{G}$, we few-shot prompt a teacher model $\mathcal{M}$ to generate a set of ordered steps, as a plan $y$ to achieve the goal. The input to $\mathcal{M}$, including instruction and few-shot examples, takes the format shown in Appendix Figure 7. Since LLMs can be sensitive to instruction, and/or few-shot examples (Perez et al., 2021; Lu et al., 2022b), we randomize the prompt by (i) manually creating a set of semantically similar instructions and each time randomly sample from the instruction set, and (ii) using different set of in-context examples for each input. We use a subset of the existing ProScript (Sakaguchi et al., 2021) and DeScript (Wanzare et al., 2016) datasets as our seed source to form in-context examples, $\mathcal{P} = \{(g_j, y_j)\}_{j=1}^M$:

$$y_i \sim \mathcal{M}(y_i|g_i, \mathcal{P})$$

The result is a pool of 140k pairs of goal and plans, $(g, y)$, generated from the teacher model.

For the constrained and counterfactual (re)planning tasks, we also obtain conditions $c$, and modified plans $y'$ from a teacher model $\mathcal{M}$ through few-shot prompting. We manually design our prompts $\mathcal{P}$ to collect natural language conditions concerning the environment the task is performed in such as Location ("the store is closed"), Equipment ("you don't have a sharp tool"), Safety ("the car breaks down") or user's specifications such as Physical Condition and Preference ("you have an injury"). For a given goal $g_i$ and plan $y_i$, we sample conditions:

$$c_i \sim \mathcal{M}(c_i|g_i, y_i, \mathcal{P})$$

Next, we few-shot prompt $\mathcal{M}$ to rewrite an initial plan $y$ for a given goal $g$ such that it satisfies the requirement of a condition $c$:

$$y_i' \sim \mathcal{M}(y_i'|g_i, y_i, c_i, \mathcal{P})$$

The prompting templates and examples of conditions are shown in Appendix Figure 8 and Table 6.

**Step 2. Automatic Data Curation** To retain high-quality data for (re)planning under the original and constrained settings, we filter out generated samples from Step 1, i.e. generated plans, conditions and counterfactuals, that are invalid or of low quality. A plan $y$ is considered invalid if it contains an *illogical order* of steps, is *off-topic* (w.r.t the goal) or *incomplete*. Whereas a modified plan $y'$ should not only satisfies these general criteria but should also adhere to the condition.

To this end, we train separate supervised critic models to judge the quality of generated samples of different types. We collect 13K human annotations of *valid vs. invalid* samples on Amazon Mechanical Turk to train a RoBERTa-Large (Liu et al., 2019a) as our critic models (see Appendix B.1 and B.2 for more details on annotation instruction and hyper-parameter tuning). All critics are binary classifiers which identify whether a tuple of either (goal, plan), (goal, plan, condition) or (goal, plan, condition, modified plan) is valid.

Naturally, there is a trade-off between dataset size and precision. Following West et al. (West et al., 2022), we test several confidence thresholds at which the critic rejects a pair and choose the best values (0.65, 0.76, 0.82)[3] according to precision-recall curves. After filtering out low quality

---

[3]These values are for plan, condition and counterfactual plans, respectively.

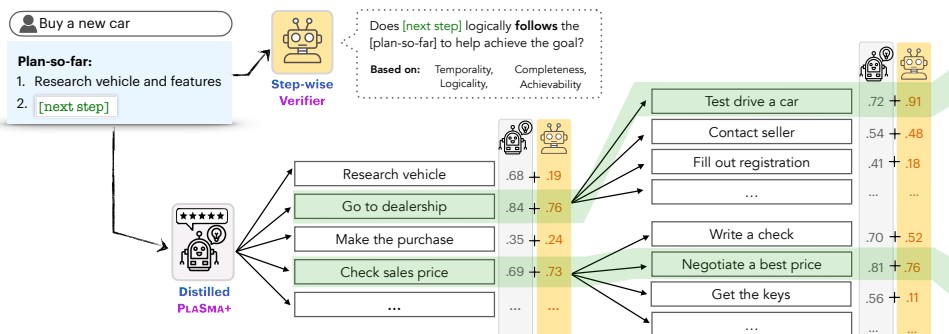

Figure 2: Verifier-guided Step-wise Beam Search. For brevity, we only showcase with $N = 5$ and $K = 2$ for the first step and $N = 4$ and $K = 2$ for the second step. The scores are for illustration.

data, our final COPLAN dataset consists of 2 main subsets including 57,794 (goal, plan) for the original **goal-based planning** task ($\mathcal{D}^{pl\cdot}$), and 43,690 (goal, plan, condition, modified plan) for the **constrained and counterfactual** settings, ($\mathcal{D}^{cp\cdot}$ and $\mathcal{D}^{cr\cdot}$). On the original planning task, COPLAN is $\times 11$ larger in scale than existing datasets (Sakaguchi et al., 2021; Wanzare et al., 2016) while keeping the precision at 74%. On the proposed constrained and counterfactual settings, our dataset is to the best of our knowledge the first large-scale constrained procedural (re)planning dataset with free-form (open vocabulary) conditions. Analyses show that the COPLAN includes a diverse array of topics covered by goals (§A.1) and conditions (§A.2).

## 2.2 PLASMA: PROCEDURAL KNOWLEDGE DISTILLATION INTO SMALL STUDENTS

After obtaining our procedural planning data COPLAN, we use it to fine-tune student models on the three different task settings described in §2.1. We consider both task-specific and multi-task distillation objectives to transfer generated procedural knowledge into the student models:

**Task-specific Distillation.** Following the common practice, we use the standard autoregressive language modeling objective (Radford et al., 2018) to fine-tune separate student models for each task:

$$\mathcal{L}(\theta) = \mathbb{E}_{(x,y) \sim D^{task}} \big[ - \log p_\theta(y | \mathcal{T}(x)) \big], \quad \text{for } \text{task} \in \{pl., cp., cr.\} \tag{1}$$

where $\mathcal{T}(x)$ is a task-specific template for each task-specific input $x$ (see right side of Figure 1).

**Multi-task Distillation.** We aim to improve the generalization of the student by exploiting the knowledge found in the three related tasks as an inductive bias (Raffel et al., 2020; Wei et al., 2022). We thus minimize the joint loss including all three task settings. We name this student PLASMA-Mul.

## 2.3 PLASMA+: ADVANCING STUDENT WITH VERIFIER-GUIDED DECODING

During inference, the student may generate logically and/or temporally ill-formed sequence of steps $\mathbf{y} = \{s_t\}_{t=1}^{T}$ as it is only trained to maximize the next-token probability. For example, in Figure 2, it may generate "write a check" at step 3 with relatively high confidence due to a spurious correlation between "sales price" and "check". We mitigate this issue via step-wise guided decoding. Rather than generating plans greedily, we instead generate step-by-step by sampling several candidate next steps and searching for those with a high log-probability under both the distilled student and a verifier. The verifier is tasked to check for sequential ordering and semantic completeness. In an embodied setting, the verifier could be taken over by any affordance or safety module (Ahn et al., 2022) that determines the executability of an action in a given environment.

**Step Verifier.** We introduce a verifier, which is trained to check the validity of plan steps and encourage PLASMA to produce more temporally and causally valid plans. The verifier takes as input a goal, the plan-so-far and a candidate next step and outputs a continuous validity score $p_{\text{verifier}}(s_t | g, s_{<t}) \in [0, 1]$.

We implement the verifier by fine-tuning a RoBERTa model (Liu et al., 2019b) to classify a candidate step as valid or invalid. For training, we reuse only 3K human-written plans from existing datasets (Sakaguchi et al., 2021) to form positive examples (valid next steps). However, since no negative

examples are readily available, we automatically create a set of invalid steps as pseudo-negative examples. Inspired by the common model errors, we design perturbations over ground-truth plans to target sequential ordering , semantic completeness , topicality, and fluency .[4] See Appendix B.3 for details on perturbation strategies. Our verifier achieves an F1 score of 78% on a held out test set.

**Verifier-guided Step-wise Beam Search.** We illustrate our *verifier-guided decoding* in Figure 2. The procedure generates a plan $\mathbf{y} = (s_1, ..., s_T)$ by sequentially sampling and pruning the next step candidate $s_t$. Concretely, at each iteration, it selects and expands a size-$K$ beam of plan-so-far, $Y_{t-1} = \{s_{<t}^k\}_{k=1}^K$, and generates $N$ next-step candidates,

$$Y_t = \cup_{s_{<t} \in Y_{t-1}} \{(s_{<t} || s_t^n) \mid s_t^n \sim q(.|\mathcal{T}(x, s_{<t})\}_{n=1}^N \qquad (2)$$

where $||$ is concatenation, $x$ is a task-specific input, and $q$ is a decoding algorithm. We encourage exploration at each step, by generating candidates using multiple decoding methods such as beam search, and nucleus sampling with temperature 1.0.

To select the top-K scoring next-step candidates $S_t^*$, we use a value function $v(s_{\leq t}) \rightarrow \mathbb{R}$ which returns the weighted sum of normalized sequence log-likelihood from the student model and the verifier validity score,

$$S_t^* = \arg \text{top-K}_{s_{\leq t} \in Y_t} v(s_{\leq t}) \qquad (3)$$

$$v(s_{\leq t}) = \alpha \log p_\theta(s_{\leq t}) + (1 - \alpha) \log p_{\text{verifier}}(s_t|g, s_{<t}) \qquad (4)$$

with $\alpha$ controlling the impact of the distilled student and the verifier. The search ends when the beam contains $K$ completed plans. We return the highest-scored plan as the final output. Our step-wise beam search strategy maintains a diverse set of candidate plans during the decoding process, allowing the model to explore multiple plausible paths before converging on a most promising one.

## 3 EXPERIMENTS

**Implementation Details.** While any model with few-shot capabilities could be used, we choose our teacher model $\mathcal{M}$ to be GPT-3 `text-curie-001` (Brown et al., 2020) for collecting the goals and initial plans, and GPT-3 `text-davinci-003` for collecting conditions and counterfactual plans.[5] We sample data points from GPT-3 using nucleus sampling ($p = 0.98$) and temperature of $T = 0.9$. For our student models, we try a range of model sizes in T5 family (Raffel et al., 2020), such as T5-large, T5-3B, and T5-11B. Student models are trained using Huggingface Transformers (Wolf et al., 2020). Main experiments can be done on 2 GPUs with 48GB of memory.

During inference, we use a beam $K = 5$ for regular beam search, and $N = 10$ (next-step candidates), beam $K = 5$, $p = 0.9$, and $\alpha = 0.5$ for our verifier-guided step-wise decoding (see §2.3).

**Baselines.** For each task, we compare our distilled students with their corresponding teacher, zero-shot and few-shot variants of GPT-3 (Brown et al., 2020), COCOGEN (Madaan et al., 2022) and human performance (when available). COCOGEN frames the planning task as a code generation task and use a pre-trained code LM (`code-davinci-002`) in a few-shot setting.

Next, we present the experimental setup for each task, along with their results.

### 3.1 GOAL-BASED PLANNING

In this section, we aim to study two key research questions through our experiments. Firstly, we seek to investigate the extent to which scale impacts the distillation of procedural knowledge. Secondly, we aim to examine whether the scale gap can be bridged through the use of multitasking and/or a novel decoding algorithm. In essence, we seek to determine whether small language models can perform procedural planning tasks with the same level of proficiency as large language models.

**Evaluation Set.** For the original planning task, we use human-written plans from the test set of `ProScript` (Sakaguchi et al., 2021) dataset as our evaluation data.

---

[4]In total, we automatically create 47K +/- pairs of (plan-so-far, next-step) using 3K human-written plans.

[5]In our preliminary experiment, we found `text-davinci-003` (the strongest GPT-3 version at the time) to be helpful for the more challenging counterfactual data collection.

**Setup.** We compare several student models of varying scales (770M-11B) with the teacher model, `text-curie-001`, and extremely large scale models (175B). For all student models, we decode using both regular beam search (PLASMA) and our verifier-guided step-wise beam search (PLASMA+).

**Metrics.** Since there may exist many equally valid plans to a goal, we conduct human evaluations for the main results and report automatic metrics such as BLEU (Papineni et al., 2002), ROUGE (Lin, 2004) and BERTScore (Zhang et al., 2020) in Appendix Table 7. We ask human annotators on the Amazon Mechanical Turk (AMT) platform to rate the generated plans for 250 randomly sampled goals on three aspects: 1) `Order`: how well-ordered the plan is (captures sequential correctness), 2) `Coverage`: how well the plan covers the necessary steps to accomplish the goal (captures semantic completeness), and 3) `Overall quality`: overall quality and correctness of the plan. Details of the human evaluation can be found in Appendix D.3 Figure 10.

Table 1 and Figure 3 summarize the human evaluation for the original planning task.

**Does scale matter?** Larger models perform relatively better across all aspects.

**Does multi-task distillation help bridge the scale gap?** As we observe, multi-task distillation almost always wins over its task-specific counterpart with the exception of the smallest student, PLASMA (770M). We posit that very small student models might not have enough capacity to leverage the related tasks efficiently during multitasking.

**Does verifier-guided decoding help bridge the scale gap?** Pairing models with our verifier-guided step-wise decoding substantially improves performance across students of varying sizes over all aspects. Specifically, compared with regular beam search, our proposed decoding results in 7%48% relative improvements in overall quality across different student sizes. The improvements achieved by the proposed decoding is larger for smaller students. We showcase the comparisons with qualitative examples in Table 8.

The best distilled students with 770M, 3B, and 11B parameters achieved respectively 14.13%, 16%, and 22.59% relative improve-

Table 1: Averaged 5-point Likert scale human evaluation for the goal-based planning. Small students paired with our decoding algorithm consistently outperform their teacher (`text-curie-001`) and are competitive with order of magnitude larger models in zero/few-shot settings. *CoCoGen (Madaan et al., 2022) is a 16-shot baseline using code LLM.

| Model$_{size}$ | | Coverage | Order | Overall Quality |
|---|---|---|---|---|
| **Distilled 770M** | PLASMA | 3.18 | 3.64 | 3.17 |
| | PLASMA+ | 4.25 | 4.55 | 4.28 |
| | PLASMA-Mul | 2.84 | 3.36 | 2.85 |
| | PLASMA-Mul+ | 4.16 | 4.48 | 4.23 |
| **Distilled 3B** | PLASMA | 3.78 | 4.07 | 3.83 |
| | PLASMA+ | 4.38 | 4.60 | 4.35 |
| | PLASMA-Mul | 3.96 | 4.35 | 4.03 |
| | PLASMA-Mul+ | 4.29 | 4.62 | 4.33 |
| **Distilled 11B** | PLASMA | 4.01 | 4.33 | 4.03 |
| | PLASMA+ | 4.33 | 4.60 | 4.39 |
| | PLASMA-Mul | 4.24 | 4.59 | 4.28 |
| | PLASMA-Mul+ | **4.53** | **4.77** | **4.58** |
| **Curie (Teacher)** | few-shot (5) | 3.75 | 4.27 | 3.75 |
| **Davinci (175B)** | zero-shot | 4.83 | 4.87 | 4.84 |
| | few-shot (5) | **4.88** | **4.90** | **4.90** |
| **COCOGEN (175B)** | few-shot (16) | 4.48 | 4.70 | 4.55 |
| **Human** | | 4.56 | 4.61 | 4.57 |

ments over their teacher model (`text-curie-001`). Finally, our best distilled model (11B PLASMA-Mul+) performs equally well as human and is competitive with orders-of-magnitude larger models (175B).[6] These results support our claim that a smaller model can, in fact, be as powerful as larger models when augmented with smarter decoding-time techniques. Figure 3 visualizes how we bridge the scale gap using our multi-task distillation and verifier-guided decoding. Since the initial submission, we conduct an additional comparison with GPT-4 (see Table 14), indicating similar trends.

**Effect of symbolic distillation.** In this experiment, we investigate the utility of CoPlan that is obtained through symbolic distillation in the presence of manually curated `ProScript` dataset (Sakaguchi et al., 2021). We thus compare a T5-11B distilled model trained on CoPlan with a T5-11B model trained only on `ProScript`, and the mix of both. Due to potential distribution shifts, we evaluated them on both their in- and out-of-domain test sets. We generate plans using our proposed verifier-guided decoding for randomly sampled 150 goals from `ProScript` and COPLAN. We use the same human evaluation setup as before. Table 2 shows that training on our LLM-generated

---

[6]Pairwise annotator agreements (i.e., how often do two annotators agree on the answer) are 0.78, 0.84, and 0.80 for coverage, order and overall quality, respectively.

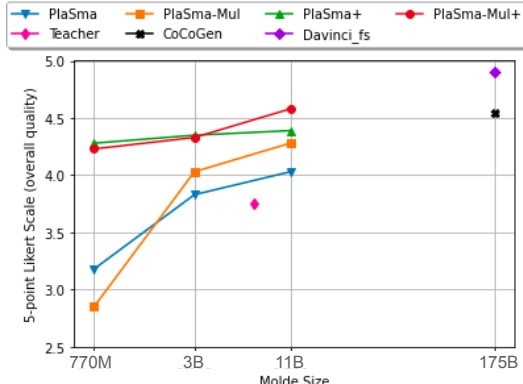

Table 2: Effect of symbolic knowledge distillation. The model trained on our CoPLAN dataset transfers better to other dataset, ProScript.

| Test on → | ProScript | | | CoPLAN | | |
|---|---|---|---|---|---|---|
| Train on ↓ | Coverage | Order | Overall Quality | Coverage | Order | Overall Quality |
| ProScript | 4.47 | 4.68 | 4.51 | 4.51 | 4.81 | 4.58 |
| CoPLAN | 4.58 | 4.78 | 4.73 | 4.72 | 4.86 | 4.73 |
| Mix | **4.82** | **4.83** | **4.83** | **4.77** | **4.88** | **4.78** |

Figure 3: Bridging the scale gap in goal-based planning. Smaller models are able to achieve comparable performance and sometimes surpass larger models via multi-tasking and guided decoding.

CoPLAN dataset, consistently transfers better to human-written dataset, ProScript across all dimensions. Training on the mix of both datasets, however, achieves the best performance.

## 3.2 CONSTRAINED AND COUNTERFACTUAL (RE)PLANNING

Here, we seek to benchmark language models' planning abilities under constrained (contextually grounded) situations. This task goes beyond the original planning task, requiring models to produce novel linguistic alternatives to unseen situations.

**Evaluation Set.** We created the test set of CoPLAN by generating conditions and counterfactual plans for the human-written (goal, plan) in the ProScript. Additionally, instead of using trained critic to filter out low-quality samples, we used human annotators to verify them. We only used human-verified tuples of (goal, plan, condition, cf. plan) as the test set of CoPLAN.

**Setup.** We compare 3B and 11B student models with GPT-3 Curie and text-davinci-003, the 175B teacher, in zero/few-shot settings. During inference, we use our verifier-guided step-wise decoding with $\alpha = 0.75$ to outweigh student model's probability over the verifier validity score.[7]

**Metric.** We conduct human evaluation on the AMT. We generate (counterfactual) plans for 300 randomly sampled examples using each model. We ask 3 workers to rate if each generated plan contains the necessary steps to make the goal achievable *while satisfying the condition*. We provide 3 answer options: **A**: The plan contains all the necessary steps to meet the requirements of the condition on the goal, **B**: The plan addresses the condition, but it is trivial and lacks thoughtfulness[8], and **C**: The plan does NOT address the condition or does so very poorly. We take the majority vote for the final results. Details on crowd-sourcing human evaluation can be found in Appendix Figure 12.

**Results.** Figure 4 depicts the results. Large students perform better on both tasks. In constrained planning, our 11B PLASMA-Mul+ demonstrates a 93.33% success rate in producing high-quality plans while adhering to the given condition, which is comparable to the performance of the 175B parameter Davinci model in a zero-shot setting. Furthermore, our model generates slightly fewer low-quality plans, only 7 as opposed to 12 by Davinci. While multi-tasking seems to be somewhat helpful in constrained planning, this is not always the case for replanning. We hypothesize that the reason for this could be that the original and constrained planning tasks, which do not involve modifying an existing plan, may negatively impact the replanning task. The best performance for the counterfactual replanning is achieved by Davinci (90%) followed by PLASMA+ (86.33%).[9] Nonetheless, statistical $T$-test of our best models for constrained and counterfactual (re)planning tasks indicate that they are statistically on par with the much larger Davinci GPT-3.5 (175B). Human-annotated error types are reported in Appendix Table 11, showing "missing necessary steps" is the most prevalent mistake.[10]

---

[7]We performed a hyperparameter search over $\alpha = \{0.5, 0.75, 0.8\}$.

[8]Example: addressing the condition "you have no money" with adding a step "find money" in the plan.

[9]Pairwise annotator agreements are 0.96 and 0.94 for constrained and counterfactual (re)planning.

[10]Results with 95% confidence intervals are reported in the Appendix Table 13 and Figure 9.

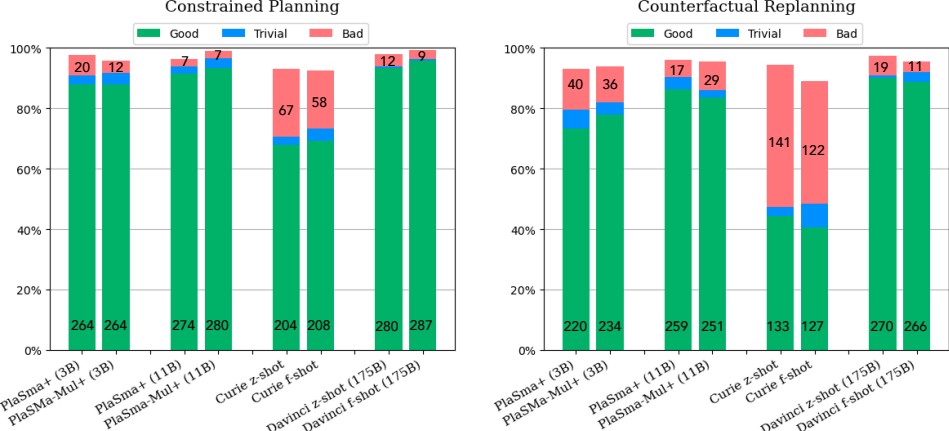

Figure 4: Human evaluation of 300 generations. PLASMA+ models (in left and right plots) are trained on the constrained and counterfactual (re)planning subsets of COPLAN. Statistical $T$-test indicates that our best models for constrained and counterfactual (re)planning are statistically on par with the much larger Davinci (175B) and are able to generate good plans 93.33% and 86.33% of the times.

We provide qualitative examples of model generations across all three tasks in Table 4. More examples of (good and bad) generations according to human annotators are provided in Appendix Tables 9, 10.

## 3.3 APPLICATION TO EMBODIED AGENTS

As an extrinsic evaluation, we investigate the application of PLASMA in a domain with hard executability conditions. We evaluate PLASMA on the task of planning in the VirtualHome (Puig et al., 2018) environment. In this environment, agents can perform household activities, e.g. "paint ceiling", through programs, in the form of supported actions (42 in total) and arguments. For evaluation, we use their test set consisting of 88 goals (and corresponding gold programs). We compare our best student PLASMA-Mul (11B) with the best-performing model on VirtualHome environment according to Huang et al. (2022). Specifically, we compare with Planner, a 1-shot GPT-3 (175B)

Table 3: Human-evaluated correctness along with (automatic) executability and Longest-common subsequence (LCS) scores on VirtualHome (Puig et al., 2018). Steps generated by our model are more executable and correct for accomplishing the task.

| model | Executability | LCS | Correctness |
|---|---|---|---|
| | (%) | (%) | (%) |
| Planner (175B) (Huang et al., 2022) | 77.17 | 19.10 | 18.33 |
| PLASMA-Mul$^{FT}$ (11B) | 76.38 | 28.36 | 41.38 |
| PLASMA-Mul+$^{FT}$ (11B) | **94.18** | **31.93** | **43.68** |
| Human | 100 | N/A | 66.66 |

model with several inference-time strategies designed to ensure executability in embodied environments. Following their setup, we translate generated steps from natural language to steps executable in the environment. To apply our model to VirtualHome, we finetune PLASMA-Mul on $\sim$ 4K human labeled examples and also finetune the verifier on the same data using the method described in Section 2.3. This human-labeled data, obtained from previous work (Huang et al., 2022), consists of pairs of ⟨task, natural language plan⟩. The same data was used to finetune supervised models including GPT-3 13B, resulting in performance inferior to that of Planner (Huang et al., 2022).

We evaluate models on: (i) **Executability**: measures whether a generated action is executable within the environment, (ii) **Correctness**: measures whether the generated actions accomplish the task, and (iii) Longest common subsequence (**LCS**): measures the action-level similarity between a generated program and a human-annotated one. We show, in Table 3, that PLASMA generates steps that are significantly more executable (according to automatic metric) and also more correct/complete (according to human judges). This suggests successful application of PLASMA in embodied setting. More experimental details can be found in Appendix E.

Table 4: PLASMA generations for *Constrained planning* and *Counterfactual replanning* tasks. More examples are provided in Appendix Table 15.

| Goal | Condition | Initial Plan | Generated (Counterfactual) Plan |
|---|---|---|---|
| Empty lint filter | | | Find lint filter > Remove lint filter > Clean lint filter > Replace lint filter |
| | If the lint trap is too hot to touch | | Unplug dryer > Open dryer > Locate lint trap > Use heat-resistant glove or mitt to pick up lint trap > Pull lint trap out of dryer > Empty lint trap |
| | | Load clothes into dryer > Locate lint trap > Move hand towards lint trap > Grip lint trap with fingers > Pull lint trap out of dryer > Use other hand to lift lint from lint trap > Empty lint filter | Load clothes into dryer > Locate lint trap > Use gloved hand to move hand toward lint trap > Grip lint trap with fingers > Use other hand to lift lint from lint trap > Remove lint trap from dryer |

## 4 RELATED WORKS

**Procedural Planning** The problem of planning to accomplish a goal via sub-steps is widely studied in two contexts. One is script knowledge generation, which is a long-standing NLP problem (Schank & Abelson, 1975a). Collecting script knowledge requires either human annotation (Wanzare et al., 2016), unsupervised feature-based extraction (Chambers & Jurafsky, 2008), or, more recently, methods that utilize task-specific fine-tuned LLMs (Sakaguchi et al., 2021) and pipeline-based approaches (Sancheti & Rudinger, 2022). In addition, there is a line of procedural planning that involves planning with executable actions that can be executed by robots in real-life environments (Huang et al., 2022; Ahn et al., 2022; Wu et al., 2022; Jansen, 2020; Guan et al., 2023). Recent approaches view planning as a conditional text generation problem using LLMs (Madaan et al., 2022; Huang et al., 2022; Ahn et al., 2022; Lu et al., 2023). Despite showing strong performance, their success heavily relies on scale.

**Symbolic Knowledge Distillation** Crowd-sourcing human-written datasets at scale is both challenging and costly, leading to a growing interest in using LLM-generated data to train smaller models which falls under the conceptual framework of symbolic knowledge distillation (West et al., 2022). In a concurrent work, Yuan et al. (Yuan et al., 2023) proposed a similar approach to distill script knowledge from LLMs for constrained planning task. However unlike our conditions which allows nuanced and free-form format, their constraints are limited to specific types by extending an original goal with a modifier. Relatedly, Collins et al. (2022) benchmarked LLMs' planning abilities (Valmeekam et al., 2023) under 28 manually constructed constrained goals. We instead investigate a broader range of constraints in a larger-scale COPLAN and distill this knowledge into smaller models.

**Decoding-time Algorithm** Decoding-time algorithm is an emerging approach for adapting language models' output for task-specific characteristics. Works in this line often focus on incorporating explicit lexical constraints (Lu et al., 2021; 2022a; Hokamp & Liu, 2017; Pascual et al., 2020). Besides discrete lexical constraints, applying continuous optimization functions, e.g. KL loss, has been found to be effective (Qin et al., 2020; 2022; Kumar et al., 2021; Hoang et al., 2017). Perhaps our approach is most similar to function-guided decoding methods. Krause et al. (Krause et al., 2021) and Yang et al. (Yang & Klein, 2021) fuse next-token probability with desired attributes' probabilities at inference using a discriminator model. These and related token-level beam search variants assume access to per-token logits and gradient updates. Our decoding method however only relies on model log-probabilities and a verifier to facilitate semantic and temporal constraints at a step level.

## 5 CONCLUSIONS AND FUTURE WORK

In this paper, we focus on procedural planning, a challenging task that involves decomposing high-level goals into ordered steps. We introduce PLASMA as an effective approach that uses smaller and more accessible models. By leveraging symbolic procedural knowledge distillation and an inference-time algorithm, we have endowed smaller models with enhanced procedural knowledge and planning capabilities. Furthermore, we introduced the task of Counterfactual Planning, which involves generating/revising plans to accommodate realistic counterfactual scenarios. Our results demonstrate that significantly smaller models can effectively compete with and often outperform their larger teacher models in both original and counterfactual settings. We hope our work sheds light on new directions towards developing smaller yet powerful multi-modal models for (counterfactual) procedural planning and reasoning.

## ACKNOWLEDGEMENTS

This work was funded in part by the DARPA MCS program through NIWC Pacific (N66001-19-2-4031), and the Allen Institute for AI. We thank the Beaker Team at the Allen Institute for AI for helping with the compute infrastructure, OpenAI for providing access to the GPT-3 API, and the anonymous reviewers for the helpful discussions.

## ETHICS STATEMENT

### IRB AND ANNOTATION ETHICS

We obtained IRB exemption for our data collection and evaluation from our institution's internal review board. In full compliance to the exemption clauses as published in the code of federal regulations (45 CFR 46.104(d)(2,3)), we did not collect any deanomyzing information, and we do not publish our dataset with worker specific information such as the MTurk's worker id. Based on our exempted status, according to our internal regulations, does not require for us to use consent forms with our crowdsourcing.

Additionally, our data collection and evaluation efforts only involve human judgments about world knowledge relating to general real-world goals and plans. We have no reason to believe that our crowdsourcing posed harm or discomfort beyond the minimal risk as defined by 45 CFR 46.102(i).

### LIMITATIONS

One potential limitation of our work is that the verbalization component of our framework involves open text generation from large-scale language models (GPTs). Works such as Bender et al. (Bender et al., 2021) have argued that generations from LLMs can be prone to harmful biases stemming from the massive language data they are trained on. In the process of constructing the dataset, we have not directly observed levels of biases to cause us alarm. We believe harmful and discriminatory generations are largely mitigated by the very nature of the goals and scripts we obtain: our data is primarily composed of low-level everyday situations such as education, self-care, and mundane chores like vacuuming the floor or cooking a meal (see §A.1, A.2). This said, we acknowledge that prejudices like gender roles, for example, do also surface in the most mundane scenarios.

A related limitation is that LLMs have been trained on primarily English pretraining data, likely sourced from texts that reflect North American or European culture or norms. Consequently, we note that the goals in COPLAN may reflect the goals that are most culturally expected or appropriate to the cultures of English-speaking countries. This is also expected of the plans that may include culturally limited processes and procedures. This should be a consideration that any follow-up studies using our data and model should attend to. Extending our study to include more socio-culturally inclusive goals and plans is a compelling direction for our future research.

### BROADER IMPACTS

Related to the concerns discussed in the Limitations section above, it is important for any downstream application to be aware that our data may have a limited representation of the goals and procedures of dominant cultures of English-speaking countries.

## REPRODUCIBILITY STATEMENT

We include all experimental details for reproducing the distillation and decoding algorithm in the beginning of §3, Appendix B, and E. Additionally, instruction for collecting COPLAN and human evaluations are provided in §2.1 and Appendix D.3.

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

Supplementary Material

## A    CoPlan Analysis Details

### A.1    Goal diversity

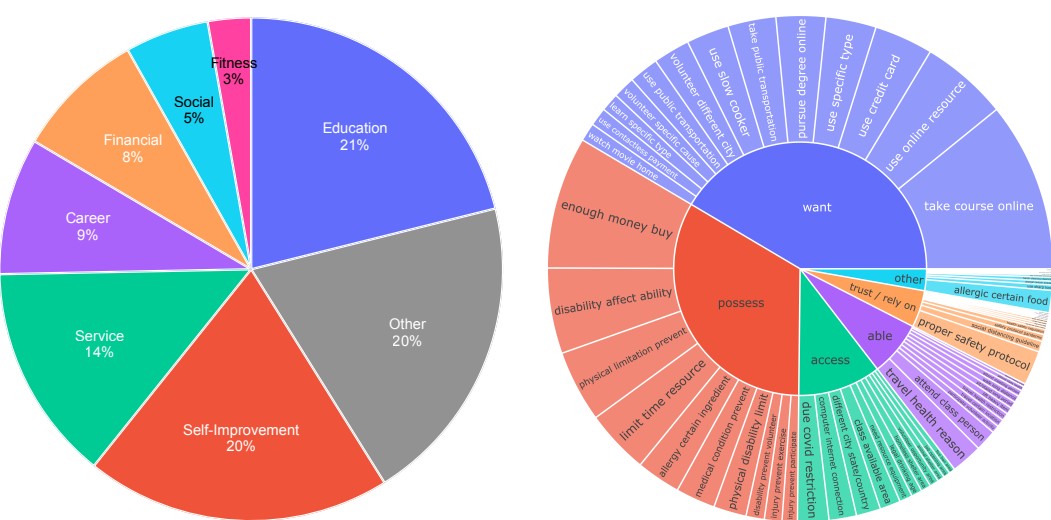

Figure 5: Goal diversity in CoPlan     Figure 6: Condition diversity in CoPlan

In this section, we demonstrate that the goals in our CoPlan dataset broadly cover a diverse set of everyday, real-world human activities.

For this analysis, we first define seven goal-relevant categories based on categories defined by the US Bureau of Labor Statistics[11]: (1) **career** and work related activities; (2) **education** and professional growth; (3) **financial** and commercial activities; (4) **fitness** and health; (5) **service** and civic activities; (6) **social** activities and relationships; and (7) **self-improvement** and leisure.

Next, using the seven categories, we manually annotate 200 most frequent verb unigrams, 300 most frequent noun unigrams, and 300 most frequent nominal (nouns + adjectives) bigrams extracted from the goals statement. Only when the unigram (e.g. "make") or the bigram (e.g. "new word") indicates one of the seven categories (e.g., "close friend" for relationship or "college university" for education) the instance is annotated with the category. Otherwise, it is annotated with an eight category, **other**. For each goal in CoPlan, each (verb, noun) unigram or (nominal) bigram casts a category as a vote if found in the annotated data. If not found, then it casts other as vote. Majority vote is taken as the category of the larger goal statement.

Figure 5 shows the distribution of the activity types in CoPlan. Education is the largest category ("join an online course to learn a new language") followed by self-improvement ("develop my creative writing skills"). Service ("cooking meals for a homeless shelter"), career ("get interview for a new job"), and financial ("upgrade to a new car") are the next largest categories. The other category includes miscellaneous activies like chores and events like "vaccuum the livingroom floor".

### A.2    Condition diversity

We assess the diversity of the conditions in CoPlan by analyzing the verbal use and nominal trigrams employed in the statements.

We manually analyze 20 most frequent verbs and phrasal verbs (e.g., "have access") appearing in the condition statements. The verbs are grouped into 5 semantic categories: (1) **want** (to want, to desire, etc); (2) **possess** (to have, to possess, etc); (3) **access** (to obtain, to get, to procure etc); (4)

---

[11]https://www.bls.gov/news.release/atus.t12.htm defines 11 categories to cover common everyday civilian activities. We cluster these categories into five.

**able** (to be able to, be capable of, etc); and (5) **trust** (to be safe, to rely, etc). Note that each of these categories include conditions of both polarity; for example, for **possess**, it includes both the condition imposed by having ("have enough money") and by lacking ("not have enough money"). A sixth category, **other**, was included for the verbs not included in the above categories. For each condition in COPLAN, the first trigram made up of verbs, adjectives, and nouns appearing after the main verb (e.g., "If you *want* to [apply to an online program]" –> main verb: *want*, trigram: *apply online program*) were extracted. Trigrams were then associated with each of the 5 semantic categories based on the main verb.

Figure 6 shows the most frequent unique trigrams in each category. The graph includes the 20 most frequent trigrams for each category. The displayed trigrams were manually clustered when appropriate for readability purposes (e.g., "take course online" clustered with "take online course").

We find a wide variety of real-world constraints that pose varying levels of restriction such as preference and desire ("want to take an online course") and hindrances posed by the state of having or not having something ("not having enough money" or "having a disability").

## B  ADDITIONAL EXPERIMENTAL DETAILS

### B.1  CRITIC MODELS: COLLECTING HUMAN ANNOTATIONS

We gather human annotations of *valid vs. invalid* teacher generations. Annotations are crowdsourced through the Amazon Mechanical Turk (AMT) platform. We qualify 263 best performing workers through a paid qualification round. Additionally, we chose annotators among those who were located in US, GB and CA, and had 98% approval rate for at least 10,000 previous annotations. Crowdworker compensation for qualification and annotation HITs is maintained at an average of $15 per hour.

**Plans.**  For plans, the crowdworkers were presented with randomly-sampled 13K generated (goal, plan) pairs, and were asked to evaluate the plans along three dimensions: *topicality*—the topic of the plan is relevant and appropriate for the goal, *ordering*—the steps in the plan are appropriately ordered, and *completeness*—the plan provides complete and informative steps to achieve the goal. We asked the workers to evaluate the goal's *achievability* as a separate (fourth) dimension. Each dimension was rated on a 5-point likert scale with three *valid* labels (Definitely, Mostly, and Somewhat; numeric value 1) and two *invalid* labels (Hardly, Not at all; numeric value 0). Each (goal, plan) pairs were annotated by three crowdworkers. The template used is shown in Figure 10.

We determine the validity of a (goal, plan) pair in the following manner. We then calculate the mean score (over the three annotator responses) for each of the dimensions. A (goal, plan) pair is considered *valid* only if: (1) it receives a score greater than $0.25$ for each of the *achievablility*, *topicality*, or *ordering* dimensions, and (2) receives a scores greater or equal to $0.65$ on the *completeness* dimension. Failing that, a pair is considered *invalid*.

**Conditions.**  For conditions, we collect human judgements on whether the condition makes the goal more specific or harder to achieve (but not impossible) on a randomly-sampled set of 6100 generated tuples of (goal, plan, condition). We include screenshot of our annotation template in Figure 11.

**Counterfactual Plans.**  And finally, for counterfactual plans, we collect 10.5K human judgements on whether the modified plan contain all the necessary steps to make the goal achievable while adhering to the condition. We include screenshot of our annotation template in Figure 12.

|  | batch size | learning rate |
|---|---|---|
| **Plan Critic** | 16 | 1e-6 |
| **Condition Critic** | 32 | 1e-5 |
| **Counterfactual Critic** | 32 | 1e-6 |

Table 5: Hyper-parameter values for training different critic models.

### B.2  CRITIC MODELS: TRAINING DETAILS

We train 3 binary classifiers (critics) for filtering out low quality teacher generations in §2.1 using pre-trained RoBERTa-Large Liu et al. (2019a). We conduct a small grid search on validation loss for batch size $bs = \{16, 32, 64\}$ and learning rate $lr = \{1e-4, 1e-5, 1e-6, 5e-6\}$. We report the effective hyper-parameters for each critic in Table 5. We use early stopping on validation loss.

### B.3 TRAINING THE VERIFIER

**Constructing Pseudo-negative Examples.** For training the step verifier, we use the human-written plans Sakaguchi et al. (2021) to construct positive examples of (plan-so-far, next-step) pairs and devise three main perturbation strategies to automatically construct negative examples as explained below:

- **Reordered Steps**: Conflicting logical order results from inaccurate causal or temporal dependencies in a plan. Thus, we apply both *near* and *distant* reordering by randomly reordering two consecutive and two distant steps.
- **Repetitive Steps**: Degeneration i.e., generating repetitive text is commonly observed in language models. Similarly, we include both *near* and *distant* repetition by repeating the immediate previous step and distant previous step as a pseudo-negative next-step.
- **Missing Steps**: Another common mistake made by language models is missing necessary steps, leading to incoherent plans. To simulate this behaviour, we randomly select a non-immediate step as a pseudo-negative next-step.

We collect a training set of 47k positive and negative pairs of (plan-so-far, next-step) using only 3k human-written plans.

**Training Details.** We fine-tune RoBERTa Large Liu et al. (2019a) as a binary classifier identifying the validity of a candidate next-step. We train for 10 epochs with early stopping on validation accuracy using batch size of 32 and learning rate of $1e-5$.

| Category | Goal | Condition |
|---|---|---|
| **Location** | Purchase gardening supplies | there are no local gardening stores nearby |
| | Sing the lyrics | you want to sing the lyrics in a recording studio |
| **Equipment** | Studying for the exam | you want to use a laptop or computer |
| | Practice pottery techniques | you don't have the right tools or clay |
| **Safety** | Take out several plates | the plates are too heavy or fragile |
| | Transport materials home | the car breaks down or runs out of gas |
| **User's condition/ specification** | Practice playing the instrument | you are unable to read sheet music |
| | Rent rock climbing equipment | you need size specific equipment |

Table 6: Examples for different categories of conditions in COPLAN dataset.

## C OUT-OF-DOMAIN EVALUATION

Collins et al. (2022) proposed two out-of-distribution reasoning tasks to evaluate LLMs, one of which involved constrained planning. For a given goal and one or more conditions, the task is to generate a plan. We evaluate PLASMA on the 28 constrained goals provided by the paper. We compare our generations to the GPT-generated plans provided by the paper and `text-davinci-002` prompted in a zero shot manner. To evaluate the generations we perform a human evaluation, as described in §D.3.

| Model | % good |
|---|---|
| PLASMA | **71** |
| GPT-3 (from (Collins et al., 2022)) | 36 |
| GPT-3 zero shot | 64 |

Table 12: Percent of generated counterfactual plans which have been rated as *good* by annotators.

The human evaluation results in Table 12 show that PLASMA outperforms the other baselines in this out-of-domain subset of counterfactual planning task.

## D EVALUATION DETAILS

### D.1 AUTOMATIC EVALUATION

We report automatic evaluation of models for the original planning task in Table 7. Note that human-written plans are not the only possible plans, hence these automatic metrics may not provide an

| model$_{size}$ | | BLEU | ROUGE-2 | ROUGE-L | BERT-f1 |
|---|---|---|---|---|---|
| **Distilled 770M** | PLASMA | 12.97 | 14.02 | 28.23 | 84.31 |
| | PLASMA + | 14.26 | 16.31 | 31.02 | 85.30 |
| | PLASMA-Mul | 14.47 | 14.43 | 27.99 | 84.02 |
| | PLASMA-Mul+ | 14.49 | 16.70 | 31.49 | 85.35 |
| **Distilled 3B** | PLASMA | 12.89 | 14.39 | 28.57 | 84.70 |
| | PLASMA + | 13.92 | 15.56 | 30.83 | 85.19 |
| | PLASMA-Mul | 13.62 | 15.42 | 29.31 | 84.80 |
| | PLASMA-Mul+ | 14.96 | 16.80 | 31.97 | 85.28 |
| **Distilled 11B** | PLASMA | 12.64 | 13.93 | 28.14 | 84.56 |
| | PLASMA + | 14.65 | 15.84 | 31.04 | 85.33 |
| | PLASMA-Mul | 13.61 | 15.67 | 29.99 | 85.10 |
| | PLASMA-Mul+ | 15.54 | 16.76 | 31.98 | 85.37 |
| **Curie (Teacher)** | few-shot (3-5) | 7.13 | 9.24 | 22.78 | 83.08 |
| **Davinci (175B)** | zero-shot | 4.98 | 7.81 | 21.38 | 81.20 |
| | few-shot (3-5) | 10.27 | 10.27 | 24.57 | 83.48 |
| **CoCoGen (175B)** | few-shot (16) | 18.07 | 21.01 | 33.99 | 86.41 |

Table 7: Automatic Evaluation Results for the Planning task. Note that the human-written plans are not the only possible plans, hence these automatic metrics may not provide an informative measure of performance.

informative measure of performance. To further verify this, we computed the correlation between the most commonly used BLEU score and human scores. We find that BLEU has very weak correlations to human scores of coverage, ordering an overall quality, with a Pearson correlation of 7.7%, 5.9%, and 5.6%.

### D.2 CONFIDENCE INTERVALS

We provide the 95% confidence intervals for our main results on goal-based (Table 13) and constrained and counterfactual (re)planning (Figure 9).

### D.3 HUMAN EVALUATION ON AMT

All human evaluations were conducted on the Amazon Mechanical Turk (AMT). We sourced our annotators from the same pool of qualified workers (see B.1). Throughout the entirety project, we maintained an average of $15/hour pay rate based on our estimation of time needed to complete the task. Each examples were annotated by 3 workers and majority vote was taken for the reported results. The layout templates for evaluating plans and counterfactual plans are shown in Figures 10 and 12, respectively.

## E EXPERIMENTAL DETAILS OF VIRTUALHOME EVALUATION

We follow the same experimental setup and metrics for evaluation as Planner Huang et al. (2022). The test set consists of 88 high-level goals. To translate a generated natural language step into an executable step, we follow Huang et al. (2022) and find an executable action closest in embedding space to the generated step. To compute these embeddings, we use the `stsb-roberta-large` model. Executability and LCS are computed identical to Huang et al. (2022). Due to challenges with reproducibility of GPT-3 outputs, evaluation results of GPT-3 do not exactly match between our works.

Note that the previous method (Huang et al., 2022) involves extracting the most relevant (in-domain) human-annotated examples by matching goals (Make breakfast ⟨-⟩ Make toast) as well as sampling LLMs several times for each step and ranking them.

| Model | Plan |
|---|---|
| | **Goal: put coffee in coffee machine** |
| Curie (Teacher) | Step 1: find coffee machine Step 2: unlock coffee machine Step 3: put coffee in coffee machine Step 4: close coffee machine Step 5: wait a few minutes Step 6: coffee should be ready |
| PLASMA-Mul (11B) | Step 1: get coffee Step 2: put coffee in coffee machine Step 3: turn on coffee machine Step 4: wait for coffee to brew Step 5: drink coffee |
| PLASMA-Mul+ (11B) | Step 1: buy coffee beans Step 2: grind the coffee beans Step 3: put the ground coffee into the coffee filter Step 4: put the filter into the machine Step 5: turn the machine on Step 6: wait for the coffee to brew Step 7: enjoy your coffee |
| | **Goal: put on baseball jersey** |
| Curie (Teacher) | Step 1: get baseball jersey Step 2: put baseball jersey on Step 3: take baseball jersey off |
| PLASMA (11B) | Step 1: find a baseball jersey Step 2: put on baseball jersey Step 3: take off baseball jersey Step 4: put on another baseball jersey Step 5: take off second baseball jersey Step 6: put on third baseball jersey Step 7: put on fourth baseball jersey |
| PLASMA+ (11B) | Step 1: walk to closet Step 2: open closet Step 3: look for baseball jersey Step 4: find baseball jersey Step 5: take baseball jersey out of closet Step 6: put on baseball jersey |
| | **Goal: sit at the table with dinner date** |
| Curie (Teacher) | Step 1: text or call the person to let them know that you will be late Step 2: arrive at the restaurant on time Step 3: take a seat at the table Step 4: wait for the other person to arrive Step 5: eat dinner Step 6: say goodbye and thank you |
| PLASMA (3B) | Step 1: find dinner date Step 2: go to dinner date's house Step 3: sit at the table Step 4: talk to dinner date Step 5: eat dinner Step 6: leave dinner date's house |
| PLASMA+ (3B) | Step 1: choose a restaurant Step 2: make a reservation Step 3: drive to the restaurant Step 4: park the car Step 5: enter the restaurant Step 6: find a seat Step 7: sit at the table with dinner date |
| | **Goal: brush all teeth** |
| Curie (Teacher) | Step 1: get in the car Step 2: drive to the dental office Step 3: park the car Step 4: go inside the office Step 5: pay the bill Step 6: get a receipt Step 7: brush all teeth Step 8: leave office |
| PLASMA (770M) | Step 1: Get toothbrush Step 2: Get toothpaste Step 3: Get toothbrush Step 4: Get toothbrush Step 5: Get toothbrush Step 6: Get toothbrush Step 7: Brush all teeth |
| PLASMA+ (770M) | Step 1: Get toothbrush Step 2: Get toothpaste Step 3: Put toothpaste on toothbrush Step 4: Grab toothbrush Step 5: Hold toothbrush up to mouth Step 6: Brush all teeth |

Table 8: Qualitative examples generated by different models for the goal-based planning task.

## F  COMPARISON WITH GPT-4

It is noteworthy to mention that the planning subset of COPLAN is collected from the smaller GPT model (as teacher), i.e., `text-curie-001` which is significantly less powerful that the most recent GPT-4 model. Nonetheless, we conduct a comparison of our best goal-based PLASMA model (11B PLASMA-Mul+) with its teacher, GPT-3[12] and GPT-4 in few-shot setting on 50 instances (total of 200). Results are shown in Table 14. As we observe, the trend remains the same as in Table 1, with GPT-4 slightly surpassing its predecessor (`text-davinci-003`) only in the ordering dimension.

---

[12]`text-davinci-003`

| Type | Conditional Plan |
|---|---|
| good | **Goal**:
eat ice cream *if you have an ice cream maker*

Step 1: Decide to eat ice cream
Step 2: Locate the ice cream maker
Step 3: Turn on the ice cream maker
Step 4: Add ingredients to the ice cream maker
Step 5: Follow the instructions to churn the ice cream
Step 6: Remove the ice cream from the ice cream maker
Step 7: Put the ice cream in a bowl
Step 8: Eat ice cream |
| good | **Goal**:
brush all teeth *if you are out of toothpaste*

Step 1: Decided to brush all teeth
Step 2: Locate a store that sells toothpaste
Step 3: Purchase toothpaste
Step 4: Drive to the store
Step 5: Purchase toothpaste
Step 6: Drive back home
Step 7: Unpack the toothpaste
Step 8: Open the toothpaste
Step 9: Brush all teeth |
| bad | **Goal**:
write names on the cards *if you have a shaky hand*

Step 1: find a pen or pencil with a large tip
Step 2: hold the pen or pencil in one hand
Step 3: hold the pen or pencil in the other hand
Step 4: use a pen or pencil to write the names on the cards
Step 5: once the names are on the cards, place them in a drawer or file
Step 6: close the drawer or file
Step 7: write names on the cards |
| bad | **Goal**:
walk into town *if you have a disability that limits your mobility*

Step 1: Decide to walk into town
Step 2: Put on shoes
Step 3: Get a cane or other mobility aid if necessary
Step 4: Put the cane or other mobility aid in a backpack
Step 5: Put the backpack on
Step 6: Leave the house
Step 7: Walk to the bus stop
Step 8: Take the bus into town
Step 9: Walk into town |

Table 9: Qualitative examples of the *constrained planning* task.

| Type | Original Plan | Counterfactual Plan |
|------|---------------|---------------------|
| good | **Goal**: empty lint filter
Step 1: load clothes into dryer
Step 2: locate lint trap
Step 3: move hand towards lint trap
Step 4: grip lint trap with fingers
Step 5: pull lint trap out of dryer
Step 6: use other hand to lift lint from lint trap
Step 7: empty lint filter | **Goal**: empty lint filter *if the lint trap is too hot to touch*
Step 1: Load clothes into dryer
Step 2: Locate lint trap
Step 3: Move hand towards lint trap
Step 4: Use a heat-resistant glove to grip lint trap with fingers
Step 5: Pull lint trap out of dryer
Step 6: Use other hand to lift lint from lint trap
Step 7: Empty lint filter |
| good | **Goal**: take a shower after work
Step 1: decided to take a shower after work
Step 2: drive home from work
Step 3: go to bath room
Step 4: take off clothes
Step 5: start water to get heat
Step 6: take hot shower
Step 7: take a shower after work | **Goal**: take a shower after work *if you want to take a shower at the gym*
Step 1: Decide to take a shower after work
Step 2: Drive to gym
Step 3: Go to locker room
Step 4: Remove clothes
Step 5: Enter shower room
Step 6: Start water to get heat
Step 7: Take hot shower
Step 8: Take a shower after work |
| bad | **Goal**: empty old left overs from the fridge
Step 1: go through the house
Step 2: walk into the kitchen
Step 3: locate the refridgerator
Step 4: walk to the refridgerator
Step 5: grab the door handle
Step 6: open the refrigerator door
Step 7: empty old left overs from the fridge | **Goal**: empty old left overs from the fridge *if you have difficulty bending down*
Step 1: go through the house
Step 2: walk into the kitchen
Step 3: locate the refridgerator
Step 4: walk to the refridgerator
Step 5: grab a chair or stool
Step 6: place the chair or stool in front of the refrigerator
Step 7: use the chair or stool to open the refrigerator door
Step 8: empty old left overs from the fridge |
| bad | **Goal**: buy a new watch
Step 1: decided to buy a new watch
Step 2: search the internet for watch retailers
Step 3: pick a reliable retailer
Step 4: search the retailer site for watches
Step 5: add watch to cart
Step 6: click check out
Step 7: add payment information
Step 8: add address information
Step 9: buy a new watch | **Goal**: buy a new watch *if your payment information is compromised*
Step 1: decide to buy a new watch
Step 2: search the internet for watch retailers
Step 3: pick a reliable retailer
Step 4: search the retailer site for watches
Step 5: add watch to cart
Step 6: click check out
Step 7: add payment information
Step 8: verify payment information
Step 9: buy a new watch |

Table 10: Qualitative examples of the *counterfactual re-planning* task.

| Error Type | Constrained Planning | | | Counterfactual Replanning | | |
|---|---|---|---|---|---|---|
| | Edits Required | Missing steps | Unnecessary steps | Edits Required | Missing steps | Unnecessary steps |
| Plasma+ (3B) | 4.66 | 8.33 | 3.66 | 13.33 | 19.33 | 6.00 |
| Plasma-Mul+ (3B) | 4.33 | 7.66 | 3.66 | 10.66 | 14.66 | 4.33 |
| Plasma+ (11B) | 3.66 | 5.00 | 3.33 | 4.66 | 10.00 | 3.33 |
| Plasma-Mul+ (11B) | 3.00 | 3.33 | 3.66 | 6.00 | 11.66 | 4.66 |
| `curie-001` zero-shot | 7.00 | 27.00 | 6.66 | 26.00 | 49.33 | 13.66 |
| `curie-001` few-shot | 6.00 | 25.33 | 5.00 | 30.00 | 48.00 | 13.33 |
| `davinci-003` zero-shot | 1.33 | 6.33 | 0.66 | 5.33 | 7.33 | 2.66 |
| `davinci-003` few-shot | 1.33 | 3.00 | 0.66 | 4.33 | 8.66 | 2.66 |

Table 11: Percent of generated (counterfactual) plans with each error type. "Missing Steps" is the most common error types across all models.

| $Model_{size}$ | | Overall Quality | 95% CI |
|---|---|---|---|
| **Distilled 770M** | PLASMA | 3.17 | [2.94; 3.41] |
| | PLASMA+ | 4.28 | [4.12; 4.44] |
| | PLASMA-Mul | 2.85 | [2.60; 3.09] |
| | PLASMA-Mul+ | 4.23 | [4.05; 4.42] |
| **Distilled 3B** | PLASMA | 3.83 | [3.69; 3.97] |
| | PLASMA+ | 4.35 | [4.20; 4.50] |
| | PLASMA-Mul | 4.03 | [3.85; 4.20] |
| | PLASMA-Mul+ | 4.33 | [4.17; 4.48] |
| **Distilled 11B** | PLASMA | 4.03 | [3.85; 4.20] |
| | PLASMA+ | 4.39 | [4.23; 4.54] |
| | PLASMA-Mul | 4.28 | [4.12; 4.42] |
| | PLASMA-Mul+ | **4.58** | [4.46; 4.71] |
| **Curie (Teacher)** | few-shot (5) | 3.75 | [3.54; 3.95] |
| **Davinci (175B)** | zero-shot | 4.84 | [4.77; 4.92] |
| | few-shot (5) | **4.90** | [4.85; 4.95] |
| **COCOGEN (175B)** | few-shot (16) | 4.55 | [4.43; 4.68] |
| **Human** | | 4.57 | [4.46; 4.69] |

Table 13: Averaged 5-point human 'quality ratings' for original planning along with 95% Confidence Intervals.

## G    COMPLEXITY AND DIVERSITY ANALYSIS OF STUDIED DATASETS

We analyze the complexity of proScript and CoPlan from several dimensions:

- **Lexical diversity**: We use generally accepted measures (Gehrmann et al., 2021) to analyze the diversity of datasets. We compute 1/2/3-gram entropy and the mean segmented token type ratio (MSTTR). To establish a comparison, we compute these values for three other datasets: XSUM (extreme summarization of news articles), DialogSum (real-life scenario dialogue summarization), and TinyStories. These have been specifically picked as they are stylistically different from our goal and script setup, and they often contain longer more natural sentences. In Table 16, we observe that even though the goals and steps in our dataset are shorter, the overall lexical diversity of proScript and Coplan are comparable with other datasets. XSUM displays a higher MSTTR score, but this is likely attributed to

| | Coverage | Order | Overall Quality |
|---|---|---|---|
| PLASMA-Mul+ (11B) | 4.31 | 4.68 | 4.23 |
| Curie (teacher) | 3.53 | 4.37 | 3.58 |
| GPT-3.5 (Davinci-003) | 4.78 | 4.84 | 4.81 |
| GPT-4 | 4.78 | 5.00 | 4.81 |

Table 14: Comparison of our best goal-based PLASMA model with its teacher, GPT-3.5 and GPT-4 in fewshot setting.

the characteristics of news and more formal text. We also note that the machine-generated CoPlan exhibits slightly higher lexical diversity than the human-written proScript.

- **Perplexity of an LM**: We also report the perplexity of an off-the-shelf language model which measures the degree of uncertainty (surprise) of an LM when it generates the next tokens. The higher the perplexity, the more surprised the LM is. As we see from the last column, with the exception of XSUM, the remaining datasets exhibit comparable perplexity scores. The higher perplexity in the case of XSUM is, again, attributed to its association with the news domain, a distinct characteristic compared to the other datasets, which predominantly encompass everyday scenarios. Also note that, LMs generally have lower perplexity scores on machine-generated data (as seen for CoPlan vs proScript).

**Overlapping of the plans.** We additionally analyze the amount of overlap between the steps in the train and test set. To this end, we identify the direct noun object of a given goal (e.g., "Carry [a plate] to the kitchen") and remove it from the goal (i.e., "Carry to the kitchen") as well as all the steps in the corresponding plan (e.g., "Pick up the plate" –> "Pick up". We then concatenate each goal with individual steps (i.e., "Carry to the kitchen. Pick up.") and measure the maximum longest subsequence match of goal+step in the test set over all goal+step in the train set.

In the ProScript dataset, we find that only 4.3% of steps in the test set have exact overlap with steps in the train set. If we relax the overlap to 90% and 80% (as opposed to an exact 100% overlap), this number increases to 5.5% and 10%, respectively. If the same is computed for plan overlap (as opposed to the step level; i.e., the direct object removed goal and plan—not individual steps—are concatenated), we observe 0% overlap.

In the COPLAN dataset, commuting overlap w.r.t full train set is still in progress (due to its large scale). However, on a randomly sampled 10K instances from the train set, the numbers are 1.8%, 2%, 3.3% for exact, 90% and 80% overlap, respectively. And 0% for plan overlap.

This suggests that differences in the object taken by the verb do not necessarily mean minimal changes to the plan. Intuitively speaking, this is sensical: changes in the direct object, in the real world, should affect the way we resolve a goal. For example, what extra steps we take (e.g., stack plates, but not mugs), the final goal location of the item (e.g., kitchen sink for plates, but the fridge for apples), or the manner of carrying (e.g., glasses vs. boxes) will affect the steps we take in a plan even if the goal is constant (i.e. "Carry X to the kitchen"). This is even more pronounced in CoPlan as it contains a broad set of everyday human goals (see Appendix A) which can lead to vastly distinct plans even when the event (verb) itself is the same. While learning is learning, the differences between the path taken to "Learn to play a violin" vs. "Learn to play Monopoly" vs. "Learn to speak Spanish fluently" are non-trivially different.

## H  DISCUSSION

**On the importance of Distillation.** Finetuning LLMs requires updating models' parameters which is not only costly but often inaccessible for the broader community. Reducing the scale and cost of strong models via teacher distillation is the key to developing open-sourced LMs that are accessible to all, facilitating fine-tuning and seamless adaptation to various domains and custom use cases. Given the large-scale training dataset used for PlaSma, we hope it can serve as a foundation model that

| Goal | Condition | Initial Plan | Generated (Counterfactual) Plan |
|---|---|---|---|
| Buy a new videogame | | | Find a store that sells videogames > Go to the store > Find the videogame you want to buy > Buy the videogame > Take the videogame home > Play the videogame |
| | If the game store is too far away | | Decide to buy a new videogame > Research online game stores > Compare prices and reviews > Select the game to buy > Purchase the game online > Wait for the game to be delivered > Buy a new videogame |
| | | Decided to buy a new videogame > Decide on game > Get into car > Drive to game store > Walk into game store > Find the desired game > Buy a new videogame | Decided to buy a new videogame > Decide on game > Search for the game online > Compare prices online > Purchase the game online > Wait for the game to be delivered > Unbox the game > Play the game > Enjoy the game |
| Eat some ice cream from an ice cream shop | | | Find ice cream shop > Enter ice cream shop > Find ice cream > Choose ice cream > Eat ice cream |
| | If you have allergies to dairy products | | Decide to eat some ice cream from an ice cream shop > Find an ice cream shop > Walk into ice cream shop > Ask for a dairy-free ice cream > Pick out ice cream > Pay for ice cream > Eat some ice cream from an ice cream shop |
| | | Decided to eat some ice cream from an ice cream shop > Decide to eat ice cream > Gather the family > Drive to ice cram parlor > Get family out and lock car > Walk in ice cream shop and sit > Eat some ice cream from an ice cream shop | Decide to eat some ice cream from an ice cream shop > Research ice cream shops in your area that offer dairy-free options > Gather the family > Drive to ice cream parlor > Get family out and lock car > Walk in ice cream shop and sit > Ask the staff about their dairy-free options > Select a dairy-free option > Eat some ice cream from an ice cream shop |
| get extra exercise | | | Set a goal for how many hours a week you want to exercise > Research different types of exercise > Choose one type of exercise to start with > Make a plan to follow the exercise plan > Start exercising > Keep exercising |
| | If the fitness activity presents a risk of injury | | Decide what type of exercise you would like to do > Research the activity to make sure you understand the risks and benefits > Consult with a medical professional to ensure the activity is safe for you > Find a place to do the activity > Set a goal for how often you would like to do the activity > Start doing the activity > Monitor your progress and adjust the activity as needed > Get extra exercise |
| | | Decided to get extra exercise > Open the calendar > Identify open time slots during the week > Schedule a fitness activity during free time > Set reminder for activity on phone > Attend weekly fitness activity > Get extra exercise | Decide to get extra exercise > Open the calendar > Identify open time slots during the week > Research and select a fitness activity that presents a low risk of injury > Schedule the fitness activity during free time > Set reminder for activity on phone > Attend weekly fitness activity > Get extra exercise |
| chop the wood | | | Get a chopping block > Get a chopping knife > Chop the wood > Clean the chopping block > Put the chopping block away |
| | If you have a chainsaw | | Pick up the wood > Place the wood on the ground > Set up the chainsaw > Sharpen the chainsaw > Start the chainsaw > Cut the wood > Chop the wood |
| | | Pick up axe > Pick up a piece of wood > Check the condition of wood > Make sure wood is dry enough > Set the wood upright > Swing the axe at wood > Chop the wood | Pick up the chainsaw > Pick up a piece of wood > Check the condition of wood > Make sure wood is dry enough > Set the wood upright > Start the chainsaw > Guide the chainsaw along the wood > Chop the wood |

Table 15: Additional PLASMA generations for *constrained planning* and *counterfactual replanning* tasks.

can be quickly adapted to specific domains with minimal additional annotation (like we demonstrate by adapting Plasma to VirtualHome). Moreover, we could not augment most of the LLMs with our decoding-time algorithm due to limited access to the model's log probabilities.

| Dataset | H1 | H2 | H3 | MSTTR | Perplexity (GPT-med) |
|---|---|---|---|---|---|
| XSUM | 10.45 | 15.49 | 17.15 | 0.80 | 18.55 |
| DialogSum | 9.04 | 14.38 | 16.73 | 0.66 | 13.20 |
| TinyStories | 8.65 | 14.25 | 17.80 | 0.61 | 7.53 |
| ProScript | 8.86 | 13.67 | 15.61 | 0.59 | 12.93 |
| CoPlan | 9.27 | 15.00 | 18.05 | 0.64 | 10.05 |

Table 16: Lexical Diversity of proScript and CoPlan.

**"Symbolic" AI vs. "Symbolic" Knowledge Distillation.** We would like to draw attention to the evolving use of the term "symbolic" within the contemporary AI community, particularly in the context of natural language. It is important to note that the term "symbolic" has acquired multiple connotations, and its modern usage may differ from its original application in symbolic AI (Kambhampati et al., 2021).

In our case, "Symbolic" (in symbolic knowledge distillation) refers to human-readable textual formats (West et al., 2022) rather than the transfer of obscure/soft model weights as in standard distillation (Hinton et al., 2015).

## I  EXTENDED RELATED WORKS

**Building Smaller Models.** There is a recent line of work on building general-purpose small models for reasoning tasks such as Orca (Mukherjee et al., 2023). While our work shares a similar spirit with Orca, we find the key distinction in (1) our goal is to develop a specialized small model for procedural/counterfactual planning and replanning with potential application to an embodied domain, and (2) Orca is focused on learning from GPT-4 explanations (Chain of Thought) to improve models capabilities. Nonetheless, building specialized models on top of them can be explored in future works as we only worked on models that were accessible at the time of submission.

**Example Template:**

Given a goal write down a list of steps to achieve the goal:

Goal: take a nap on the bed
Step 1: sit on the bed for a little
Step 2: pull back the blanket
Step 3: pull back the sheet
Step 4: fluff up the pillow
Step 5: lay down on the bed
Step 6: fall asleep on the bed
Step 7: take a nap on the bed
...

Goal: hire a dog walker
Step 1:

- - - - - - - - - - - - - - - - - - - - - - - - - - - - - - - - - - - - - - - -

**Prompt Prefix Generator:**

```python
def generate_prompt_prefix():
    w1_list = ["For a given goal",  "Given a goal"]
    w2_list = ["write down", "break down into", "put down" "jot
                                    down"]
    w3_list = ["steps", "subgoals", "a list of steps", "several
                                    steps", "several subgoals", "
                                    some steps", "some small
                                    steps"]
    w4_list = ["to achieve the goal", "for achieving the goal", "to
                                    attain the goal"]

    w1 = random.sample(w1_list, 1)[0]
    w2 = random.sample(w2_list, 1)[0]
    w3 = random.sample(w3_list, 1)[0]
    w4 = random.sample(w4_list, 1)[0]

    prompt_prefix = f"{w1}, {w2} {w3} {w4}.\n\n"
    return prompt_prefix
```

Figure 7: Randomize prompt template for eliciting plans.

**Prompt Template (Conditions)**

You want to use social media. How can you do this in 7 steps?
step 1: decided to use social media; step 2: Grab the phone;
step 3: Open, Start phone; step 4: Go to app store; step 5:
Download Facebook from store; step5: Open and use
facebook; step6: use social media
What is the hindrance that might affect the plan above?
If your phone screen is cracked.

You want to plant a tomato plant. How can you do this in 7
steps?
step 1: decided to plant a tomato plant; step 2 : Go to
nursery; step 3: Purchase tomato seedling.; step 4: Purchase
potting soil and a pot.; step 5: Return to home.; step 6: Plant
seedling in soil and pot.; step 7: plant a tomato plant
What is a specification that might affect the plan above?
If you want to use compost for soil.

**...** x 3

You want to print the report. How do you do this in 7 steps?
step 1: type the edited draft; step 2: save the edited draft;
step 3: open the file menu in the word processor; step 4:
select print from the file menu; step 5: select printer settings;
step 6: send document to the printer; step 7: print the report
What is the hindrance that might affect the plan above?

**Prompt Template (Counterfactual Plan)**

You want to learn how to swim. How can you do this in 7
Steps?
Step 1: Decided to learn how to swim; Step 2: Find swimming
instructor; Step 3: Travel to pool; Step4: Meet swimming
teacher; Step 5: Practice swimming during classes; Step 6:
Review mistakes with teacher until right; Step 7: Learn how to
swim.
You want to learn how to swim. How can you do this in
several steps if you forget your swimsuit?
Step 1: Decided to learn how to swim; Step 2: Find swimming
instructor; Step 3: Travel to pool; Step4: Meet swimming
teacher; Step 5: If you have forgotten your swimsuit, ask the
instructor if it is possible to borrow one or if there is a place
where you can purchase one; Step 6: Practice swimming
during classes; Step 7: Review mistakes with teacher until
right; Step 8: Learn how to swim

**...** x 3

You want to pick up pen. How can you do this in 6 steps?
step 1: look for a pen; step 2: find a pen; step 3: walk over to
pen; step 4: extend hand out to pen; step 5: reach for pen;
step 6: pick up pen
You want to pick up pen. How can you do this in several
steps if you want to pick up the pen from a high shelf?

Figure 8: Prompt templates for acquiring Conditions and Counterfactual Plans.

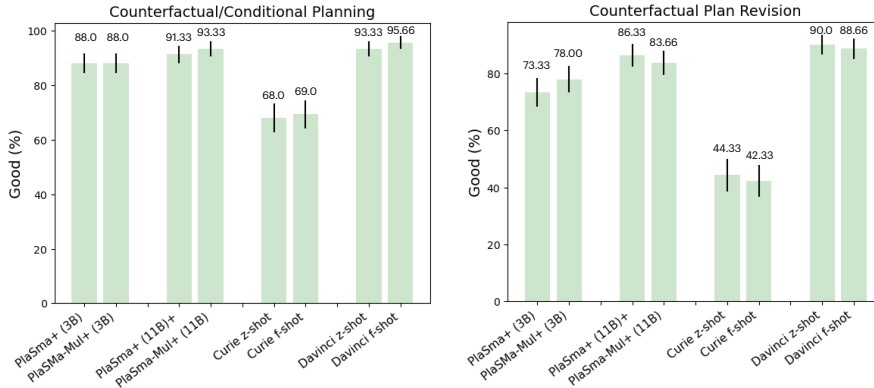

Figure 9: Human evaluation of constrained and counterfactual (re)planning tasks. We report the proportion of plans labeled as "Good" by annotators along with 95% confidence intervals. Applying statistical tests (t-test) indicates a significant difference between all PLASMA variants and Curie ($p < 0.01$) as well as a significant difference between students of different sizes ($p < 0.05$). No statistical significance was found between variants w/ or wo/ multitasking as well as between our best PLASMA and ×16 larger text-davinci-003.

**Instructions (click to expand/collapse)**

WARNING: This HIT may contain adult content. Worker discretion is advised.

Thanks for participating in this HIT!

In this HIT, imagine you are in the business of teaching people how to go about achieving everyday or life-long goals. You are handed a **goal** and a **plan** you can use to teach.

Your task is to evaluate plans based on several criteria. A good plan **doesn't contain repetitive or unnecessary steps**, is **on-topic**, **well-ordered**, and **complete**.

**Goal**   A life goal to achieve. Goal can be as simple as "cooking a dinner" to more elaborate "visiting Hawaii". It can also be fairly ambitious like "travelling to every country in the world" or time-consuming like "becoming the best sushi chef in the country".

**Plan**   The proposed plan given goal. A list of typical subgoals or steps to achieve the main goal. (e.g., for "visit Hawaii": buy ticket to Hawaii, decide what you want to see, book lodging, pack, leave for the airport)

**Grade with the following rubric.** We define the `Definitely`, `Somewhat`, and `Not at all`. Use middle values as needed.

1. **Is the plan on-topic?**:
   - `Definitely`: Topic in the plan is relevant and appropriate to the goal.
   - `Somewhat`: Topic in the plan is wanders a bit from goal, but it is okay overall.
   - `Not at all`: Topic in the plan is overall irrelevant to the goal.

2. **Is the plan well-ordered?**:
   - `Definitely`: The ordering is just fine as is.
   - `Somewhat`: I could see reordering some of these, but it would be more of a stylistic change.
   - `Not at all`: Ordering is bad or nonsensical.

3. **Is the plan complete and informative?**:
   - `Definitely`: The plan provides a complete and informative picture of what's needed to achieve the goal.
   - `Somewhat`: The steps are somewhat general, but you overall you get what you need. You might need a few more minor details.
   - `Not at all`: The plan is really bland and is dominated by unnecessary, irrelevant, and/or repetitive steps, -or- key steps are missing.

4. **Is the plan overall good?**:
   - `Definitely`: The plan is overall good. A good plan should be well-ordered, complete and contains no repititive steps.
   - `Somewhat`: The steps are somewhat general, but overall you get what you need.
   - `Not at all`: The plan is really bland and not good with repetitive steps.

NOTES:
- Steps are allowed to be general so long as the key information is there. Think: is the plan enough to give students solid grounding to start of asking relevant questions and taking relevant steps to achieve the goal?
- Note that an overall good plan should be topically relevant, ordered correctly, almost complete and contains no repetitive or unnecessary steps.
- Please do not hover too much over fine-grained differences. When in doubt, choose go with your gut instinct.
- If the goal is an incomplete thought or is nonsensical, then please choose `Not at all`

**Examples (click to expand/collapse)**

Goal: **${goal}**
Steps:
   1. ${steps_html}

|  | Definitely | Mostly | Somewhat | Hardly | Not at all |
|---|---|---|---|---|---|
| **Is plan on-topic?** | ○ | ○ | ○ | ○ | ○ |
| **Is plan well-ordered?** | ○ | ○ | ○ | ○ | ○ |
| **Is plan complete/informative?** | ○ | ○ | ○ | ○ | ○ |
| **Is plan overall good?** | ○ | ○ | ○ | ○ | ○ |

(Optional) Please let us know if anything was unclear, if you experienced any issues, or if you have any other fedback for us.

Submit

Figure 10: AMT human evaluation template for the original planning task. For validation round we substituted goal *achievability* (is the goal achievable with appropriate steps?) for *overall* question (is the plan overall good?).

Figure 11: AMT template for assessing validity of conditions for critic model training.

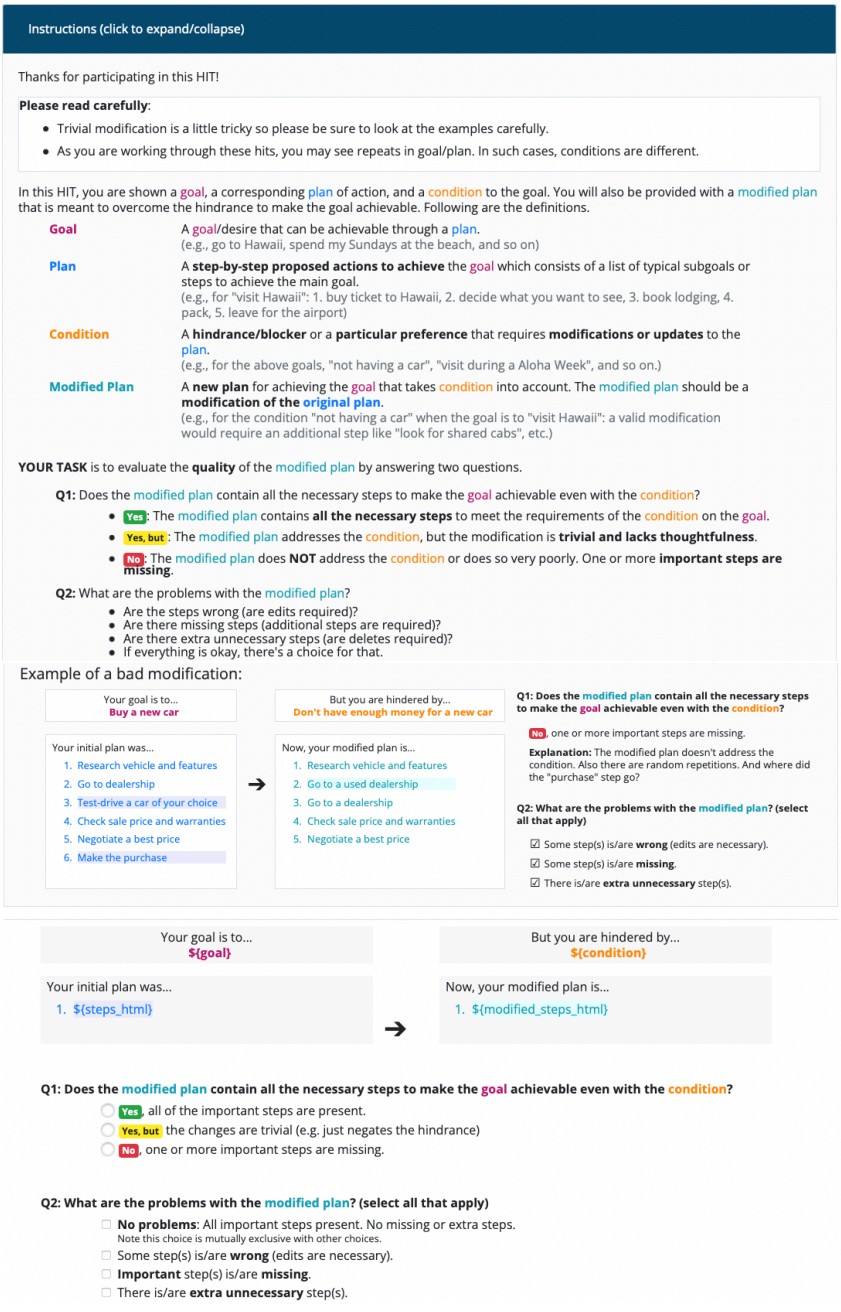

Figure 12: AMT human evaluation template for counterfactual re-planning. We use a similar layout for counterfactual planning task only removing the initial plan.

