# OpenReview forum: "PlaSma: Procedural Knowledge Models for Language-based Planning and Re-Planning"
_ICLR.cc/2024/Conference — ICLR 2024 poster_

### Official Review · Reviewer_bpqq · 2023-10-30

**Soundness:** 3 good
**Presentation:** 3 good
**Contribution:** 3 good
**Rating:** 6
**Confidence:** 4

**Summary:**

This paper proposes to leverage knowledge distillation to train smaller LMs to replicate the procedural plan generation abilities of LLMs.This serves as a cost-effective alternative to achieve the same performance as LLMs using smaller LMs. The authors generate a novel dataset, called CoPlan which includes goal-based and constrained planning tasks, as well as counterfactual replanning tasks. When trained on the CoPlan dataset, the smaller (distilled) models showcased comparable performance to their LLM counterparts. Empirical and ablation experiments further demonstrate the same.

**Strengths:**

1. Overall, the paper is well-written and has a smooth flow, which makes it easy to follow.

2. The data collection process used to generate CoPlan is novel and would be useful for researchers to collect high-quality data with minimal human involvement.

3. Some of the results shown in the paper were interesting, although not entirely surprising.

**Weaknesses:**

1. Novelty: The main contribution of this work -- to train a small LM to imitate an LLM by using the LLM as a teacher to train the small LM, is not novel. It has already been demonstrated in [A] (and has not been cited here) for a variety of tasks including complex reasoning If considered in the context of [A], the novelty here is limited to its extension to planning. The use of beam search-based planning is also very similar to [B], however, given its recency, I have discounted it in my evaluation.

2. Missing Key Experiments: While the authors motivate the use of knowledge distillation and compare their distilled models with that of the teacher model, the comparisons with the original (undistilled) model seem to be missing. Without this, it is hard to gauge the performance enhancement from distillation.

[A] Orca: Progressive Learning from Complex Explanation Traces of GPT-4, Mukherjee et al., 2023

[B] SayCanPay: Heuristic Planning with Large Language Models using Learnable Domain Knowledge, Hazra et al., 2023

--------------------------------------------------------------

Edit 1:

I'm discarding the Novelty contention given that [A] is considered "contemporaneous" according to ICLR guidelines. The authors have also answered the aspect of the missing experiment. Hence, I'm raising my score.

**Questions:**

1. How is CoPlan different from ProScript? Barring the size factor, is it the counterfactual and replanning subset that is novel? Or is there a difference in the diversity of data too?

2. Can you explain the use of the term "symbolic" in "symbolic procedural knowledge distillation"? What is "symbolic" here?

3. It would be interesting to see when distillation leads to overfitting. What factors (model size of small LM, amount of training data) does it depend on? This would help motivate the generation of a larger dataset (compared to existing ones like ProScript).

4. Minor Comment: The use of the term "task" in Sec 2.2 is ambiguous. The authors should clarify upfront that the three tasks are the three different settings that they investigate.

---

> ### Author Response · Authors · 2023-11-17
> **Response to Reviewer bpqq**
>
> We thank the reviewer for their detailed review and positive comments on our novel data collection, interesting results, and well-written paper. We are in the process of updating our pdf, meanwhile please find our response below:
>
> ### W1. Novelty and Comparison with Orca:
>
> Thanks for pointing out Orca that we overlooked. Indeed there is a recent line of work on building general-purpose small models which warrant a discussion and reference. While our work shares similar spirit with Orca, we find the key distinction in (1) our goal is to develop a specialized small model for procedural/counterfactual planning and replanning with potential application to embodied domain, and (2) Orca is focused on learning from GPT-4 explanations (Chain of Thought) to improve models capabilities. Nonetheless, building specialized models on top of them can be explored in future works as we only worked on models that were accessible at the time of submission.
>
> We will add this discussion in a new extended related work in the Appendix.
>
> Thanks for bringing [B] to our attention. We believe such contemporaneous works indicate the timeliness of our contributions as they reflect the needs and interests of the research community.
>
> ### W2. Re Missing Key Experiment:
>
> Indeed we have tried to get an undistilled T5 model to work on the planning task. However, trying several prompt formats (such as “How to <goal>?”, Provide steps to achieve the <goal>”, etc.), we couldn’t get any meaningful outputs from this baseline to run human evaluation on or apply guided decoding. Most outputs were just repeating the given inputs or empty strings. However, we successfully ran a few-shot variant of undistilled T5 (with some post-processing to format the outputs). Below is the human evaluation result for this baseline on the same 250 randomly sampled test examples (Curie and PlaSma scores are compiled from Table 1):
>
> |                         | Coverage | Order | Overall |
> |-------------------------|----------|-------|---------|
> | Fewshot T5 (undistilled) | 2.63     | 3.05  | 2.63    |
> | Fewshot Curie (teacher) | 3.75     | 4.27  | 3.75    |
> | PlaSma (best)           | 4.53     | 4.77  | 4.58    |
>
> Results of Few-shot undistilled T5 is significantly behind those of fewshot Curie (teacher) and all our model variants.
>
> ### Q1. How CoPlan is different from ProScript?
> We would like to clarify that ProScript is a small-scale **manually-curated** dataset of scripts, whereas CoPlan is (1) model generated, (2) larger scale, (3) includes counterfactual replanning, and finally (4) it broadly covers a diverse set of everyday, real-world activities both at low-level (actionable) goals and high-level long-term goals. We provided a detailed diversity analysis of goals and conditions of CoPlan in **Appendix A.1 and A.2**.
>
> ### Q2. What is "symbolic"?
> “Symbolic” (in symbolic knowledge distillation) refers to human-readable textual formats rather than the transfer of obscure/soft model weights as in standard distillation (Hinto et al. 2015). Thanks for pointing this out, we will include this explanation in the updated version.
>
> ### Q3. Re "motivation for generating larger data" and impact of "model size of small LM", "amount of training data":
>
> - Our **experiment in Table 2** investigates the utility of larger CoPlan that is obtained through symbolic distillation in the presence of manually-curated ProScript. There we compare a distilled model trained only on CoPlan with a T5-11B model trained only on proscript. Results indicate that we do benefit from a larger dataset.
>
> - We studied the impact of (student) model sizes on performance (Fig. 3 and Table 1) where we show that larger students perform relatively better but we can bridge the scale gap with multi-task distillation and our proposed verifier-guided decoding.
>
> - We also conducted experiments on the impact of training data size on test/validation losses. However, we ended up not including the latter in the paper as we thought it might not fit well with the scope. Below are the results:
>
> | Portion of Training data | Validation Loss | Test Loss |
> |--------------------------|-----------------|-----------|
> | 20%                      | 1.41            | 2.27      |
> | 40%                      | 1.34            | 2.15      |
> | 60%                      | 1.26            | 2.10      |
> | 80%                      | 1.23            | 2.06      |
> | 100                      | 1.19            | 2.01      |
>
> Based on decreased validation/test loss, we don't observe signs of overfitting. Nonetheless, a more comprehensive scaling law contribution in the context of distillation could be done in the future.
>
> We will include this in the Appendix of our updated pdf.
>
> ### Q4. Typo for the term "task"
> Thanks,  we fixed this to “three different task settings” in our updated pdf.

---

> > ### Comment · Reviewer_EJrs · 2023-11-18
> >
> > Question for the reviewer: are the references [A] and [B] accepted or they are just pre-prints?

---

> > ### Comment · Reviewer_bpqq · 2023-11-18
> > **Acknowledging author response**
> >
> > Thank you authors for taking the time to write detailed clarifications. You have answered most of my questions well, however, I'm still somewhat skeptical about the core contribution of the work. I'll update my score after discussing it with other reviewers. For now, I don't have any further questions.

---

> > > ### Author Response · Authors · 2023-11-18
> > > **Response to Reviewer bpqq**
> > >
> > > We thank the reviewer for taking the time to read our responses. We're glad that most of your questions are addressed.
> > >
> > > We'd like to summarize and highlight the contribution of our work as follows:
> > >
> > > Our novelty lies in **(1)** the design of a unified and generalizable framework for data collection, distillation, and inference-time algorithm, **(2)** proposed novel counterfactual setting in the context of language-based agents, and **(3)** the application of PlaSma as a small foundation model in embodied environments. More specifically:
> > >
> > > - We propose a unified framework for distilling procedural knowledge encoded into smaller planners with broader applications. As LLMs are gaining more attention in robotics and physically embodied applications, (1) this work can open up new **opportunities for the broader community** to explore small LMs for grounded/embodied planning **(shown empirically in Sec. 3.3)**, (2) our novel counterfactual tasks, datasets, and **enhancement of smaller LMs with counterfactual and contained (re)planning** capabilities offers a valuable and unique perspective.
> > > - Our proposed **verifier-guided decoding** differs from existing decodings in that they mostly assume access to per-token logits/gradient updates and are applied at the token level. In contrast, our method only relies on model log-probabilities and a verifier and enforces semantic and temporal constraints at a **step level**.
> > > - PlaSma achieves compelling results compared to much larger models.
> > >
> > >
> > > Once again thanks for your detailed review and for considering to update your score.

---

> ### Comment · Reviewer_bpqq · 2023-11-18
> **Good Question Reviewer EJrs**
>
> That's a good question and while I was aware of [A], I had not bothered to verify that. Orca [A] is quite old now given the fast-paced timeline (and is also somewhat popular), however, it seems the authors still haven't open-sourced the data (which also explains why I couldn't find a peer-reviewed version). Nevertheless, the framework is quite detailed in their paper, based on which there are (at least) a couple of open-sourced replications of Orca going back to June 2023 (see OpenOrca: https://huggingface.co/datasets/Open-Orca/OpenOrca and Dolphin) with impressive results. As you see, given the follow-up works, the idea of training smaller LMs to reason using knowledge from LLMs is not particularly new.
>
> As for [B], as I said, I've not taken it into account given its recency, despite its overlap.

---

> > ### Comment · Reviewer_bpqq · 2023-11-19
> > **Re: Good Question Reviewer EJrs**
> >
> > Dear SACs, ACs, Reviewers, Authors,
> >
> > Upon revisiting the reviewer guidelines, I found that ICLR considers papers as contemporaneous if it was "published" on or after May 28. Since Orca is still a preprint, and since novelty has been my main contention, I'm inclined to drop it and raise my score. The authors are still encouraged to add some comparisons in their manuscript.

---

> > > ### Author Response · Authors · 2023-11-20
> > > **Thank you (Reviewer bpqq)**
> > >
> > > We are glad that the concerns were addressed, and thank you for updating your score!
> > >
> > > Authors,

---

### Official Review · Reviewer_EJrs · 2023-11-01

**Soundness:** 3 good
**Presentation:** 4 excellent
**Contribution:** 3 good
**Rating:** 8
**Confidence:** 4

**Summary:**

The paper considers the use LLMs for generating NL instructions for a given task, called procedural plans. The paper proposes that smaller LLMs trained specifically for generation of such plans can outperform the models used as teacher, and be on par of larger models.

Algorithmically, the contributions are three:
- PlaSma: small model specialized in generating NL instructions.
- PlaSma+: uses PlaSma and an additional model for biasing the output towards higher validity. The paper refers to this bias as a “constraint”.
- CoPlan dataset

The key transversal issues are both data generation and evaluation.

The dataset (CoPlan) is generated using a combination of seed prompts, large models, and human validation.

The experimental setup is reasonable these days: use proprietary GPT as teacher and for generating data; use T5 variants as small models; BERT-variations for classification tasks.

The evaluation is more complicated. The paper reports good human evaluation results in one dataset as the plans cannot be tested. (The  appendix reports usual BLEU and ROUGE scores, perhaps for pacifying some reviewers, but for natural situations there are so many possible wordings that that might very misleading). For VirtualHome, they report an interesting success rate.

The key question is whether the smaller model is just mimicking the teacher’s behaviour. However, the paper reports that the student might outperform the teacher significantly, especially if it has enough capacity. The bias-towards verification model has a higher impact in models with lower capacity.

**Strengths:**

- Interesting problem as instructions are a key possible application of LLM. Sensible to scale and cost.
- Good description of the methodology in all aspects.
	- In particular the data generation vs curation.
- Sensible complexity of the tasks: goal, conditions, verification.
- A secondary model specialized in higher correctness is a good idea while focusing at lower capacity.

**Weaknesses:**

- The dataset ProScript is not well explained
	- It is hard to qualify the complexity of the instructions.
	- So, the results in Table 2 are hard to understand because we don’t know about the relative complexity of the task and the diversity of tasks.
- I suggest reducing the tone of the phrase “we introduce the task of counterfactual planning”. A quick search in google scholar for “plan revision” reported, for instance, these papers:
	- Ow, P. S., Smith, S. F., & Thiriez, A. Reactive plan revision. AAAI 1998.
	- Williams, K., & Burdick, J. Multi-robot boundary coverage with plan revision. In Proceedings 2006 IEEE International Conference on Robotics and Automation.
- Abuse of some terms
	- The “step verifier” is **not** verifying, but adding a bias. For instance, the paper mentions that in the case of embodied agents, that verification can be taken over by a safety module. In that scenario, with reasonable \alpha that follows the LLM when the so-called verification is not saying anything relevant, the aggregation of Eq (3) cannot prevent cathastrophic errors that are considered very attractive by the LLM. Perhaps a better name would be “quality bias” or anything saying bias.
	- Same applies to the notion of “symbolic knowledge destilation”, but we are probably too late for this one. I find it problematic that in AI we associate NL with symbolic, as the word is overloaded with a huge body of work in AI ranging from logic to graphical models. It should be more clear to call it something like “instruction distillation”.

**Questions:**

- Except for PlaSma-Mul, what does PlaSma mean when measured in different tasks. Can you elaborate on how that manifests in the experiments?
	- For instance, in Table 1, the PlaSma model is trained in the planning task, so there are precisely 6 models there. Right?
	- Those models are completely different from the ones reported in Fig 4, correct?
- Please describe the ProScript in-depth and discuss why this is a good dataset for studying this problem.
- Please discuss what other datasets could have been used, and explain why some possibilities are inconvenient.
	- This should be added to the related work section.

---

> ### Author Response · Authors · 2023-11-17
> **Response to Reviewer EJrs**
>
> We thank the reviewer for their detailed review and positive endorsement of our interesting research problem, our proposed methodology, the comprehensiveness of the task under study, and compelling results on VirtualHome. Please find our response below:
>
>
> ### Re ProScript details:
> Proscript is a manually-curated dataset of ~6k scripts at varying levels of granularity, for a wide range of everyday activities (e.g., `bake a cake`, `audition for a musical`, `go sailing`, etc.). Given the diversity and granularity of goals in ProScript, we believe it provides a good venue to study procedural knowledge in language models available through distillation.
>
> Table 2  includes both evaluation on the ProScript as well as our collected CoPlan dataset.  The goal is to investigate the utility of CoPlan that is obtained through “generation from an LLM” in the presence of manually-curated ProScript.
>
> ### Re phrase "introduce the task of counterfactual planning":
> While plan revision has been explored in classical AI (with restricted action vocabularies) and Robotics, here we are investigating counterfactual and constrained (re)planning capabilities of contemporary language-based agents. As LMs are gaining more attention in robotics and physically embodied application, it becomes imperative to enhance their capabilities by integrating counterfactual planning and contained planning mechanisms.
>
> We thank the reviewer for bringing this to our attention, and we will revise this phrase accordingly in our updated version.
>
> ### Re "step verifier" is not verifying:
> We agree with the reviewer. While during training the verifier acts as a binary classifier to identify the validity of a next-step, we are using this as a bias/regularizer term during decoding the plan (Eq. 3). Thank you for pointing this out. We will update our paper to reflect this.
>
> ### Re "Symbolic" in Symbolic Knowledge distillation:
> We totally agree with the reviewer that “symbolic” became an overloaded word in AI. However, we use this term in reference  to the previous work (West et al. 2020, “Symbolic Knowledge Distillation: from General Language Models to Commonsense Models”).
>
> ### Q1. what does PlaSma mean when measured in different tasks?
> Thank you for bringing this to our attention. Yes, except for PlaSma-Mul, the models (listed under PlaSma) in Table 1 are completely different from those in Figure 4. This distinction also applies to the models featured in the left and right plots of Figure 4. We will ensure clarity by having a different naming convention.
>
> ### Q2. Describe ProScript:
> Please see our response above. We will add this to our update pdf soon.
>
> ### Q3. Other datasets:
> For training purposes, we use our large-scale collected CoPlan dataset.
>
> For evaluation purposes, in the general domain, we use ProScript which was the only human-written dataset of this kind we were aware of. In the embodied domain, we use RobotHow/VirtualHome (Puig et al. 2018) for conducting our experiments. There are other embodied text-based environments such as Alfworld (Shridhar et al. 2020) which are similar in nature to VirtualHome. We will discuss this in our extended related work section.
>
> Note that we also used DeScript (Wanzare et al. 2016) which is a small-scale manually-written dataset as our seed examples during data generation.
>
> Puig et al. 2018. “Virtualhome: Simulating household activities via programs”
>
> Wanzare et al. ACL 2016. “A Crowdsourced Database of Event Sequence Descriptions for the Acquisition of High-quality Script Knowledge.”
>
> Shridhar et al. 2020, ALFWorld: “Aligning Text and Embodied Environments for Interactive Learning.”

---

> ### Comment · Reviewer_EJrs · 2023-11-18
> **Reject that planning in the datasets is trivial/easy**
>
> Thank you. This answer some of my questions. I’m looking forward to see the description of ProScript, including a justification of why is that a good dataset for the research question. This is a key issue: If the plans are very short, and the sample of problems is not diverse enough, the planning problem becomes a classification problem: decide which one to retrieve.
>
> If new plans are required for novel situations, but the plans are basically composing two pieces of existing plans, the problem can be solved with two classifiers. Imagine a baseline that does directly that: choose possible few segments of a plan, and then use a ranker over potential combinations. Such baseline might have a good performance in a bad dataset for testing planning algorithms. That’s why is very risky to test planning with a passive data, without a simulator. Then only alternative is that the dataset is diverse.
>
> An important factor are amount to overlapping of the plans in the train set, but also respect to the test set. In this case, even if the plans are **not symbolic**, it should be possible to compare the text of the “steps” of the plans. That’s true for both datasets.
>
> Furthermore, given the simple structure of the sentences in both datasets, the overlapping of the sentences is not enough. It’d be more informative to see how the text compare with each other by ignoring **nouns**, while keeping at least the verbs and perhaps the rest of the sentence. Why? Well, the plans for these two goals is almost the same: “carry a glass from the bedroom to the kitchen”, “carry a plate from the bedroom to the kitchen”. This actually qualifies my argument about how plans in a dataset might be easier.  If a LLM memorize the plan for glass to the kitchen, then plate to the kitchen is just editing an existing plan.
>
> Such an analysis might explain why some datasets are harder than others.
>
> I understand some papers in procedural planning are being accepted in top venues without such analysis. That’s a shame. I attribute that to a weak background in combinatorial and symbolic AI, as well as lack of awareness of the literature. We are not here to repeat those mistakes, so let’s show that the problem is interesting and cannot be solved by memorizing the plans and doing small editions.
>
> In summary,
> 1. please explain ProScript in detail,
> 2. Analyze both datasets to disprove the hypothesis zero: planning can be done just by memorizing the plans and making few combinations or adjustments.
>
> New:
> - in the main body, Please explain briefly the three tasks used for Multi-Task distillation. Then in the appendix explain the dataset in detail, and any hyper-parameter so we can reproduce the experiment. Justify any decision.
> - Please confirm that PlaSma-mul is not fine-tuned at all in the other datasets.
>
> Others:
> - the expression “original planning task” is confusing. It might be better just to say ProScript.
> - In each table where PlasSa is reported, please mention the dataset used for fine-tuning.
> - About the use of “symbolic”, I don’t think that 2020 reference is enough to use such a confusing expression, but I’m not requesting a change about that terminology. I do request to acknowledge the confusion about the term, and refer to work in symbolic AI. This paper accepted to AAAI 2022 might be useful: https://arxiv.org/abs/2109.09904
>
> PS: I haven’t read the other responses, so perhaps part of my new questions are answered there. Apologies in that case.

---

> > ### Comment · Reviewer_EJrs · 2023-11-18
> >
> > By the way, about this answer to Q3:
> >
> > > For evaluation purposes, in the general domain, we use ProScript which was the only human-written dataset of this kind we were aware of.
> >
> > These seems to imply that human-written is better. Please justify that statement. I’d argue that purely human-written might lack diversity depending on the protocol, as human annotators have incentive to write as soon as possible, and schemas for additional incentive might have limited effect.
> >
> > The key issue is not the human-writing but how the situations are generated. For instance. We can ask a human to provide some traces of the game tic-tac-toe. Or we can sample boards of tic-tac-toe and ask them to provide an answer. The later case would have more diversity of goals, but perhaps no diversity of plans, but that might be ok.
> >
> > So, when explaining ProScript, please describe the data generation as part of the argument of why ProScript is interesting to the question of the paper.
> >
> > **The contribution of using smaller models might be tainted if that’s exploiting an artifact of the datasets. So, the dataset description and analysis is crucial to prove the point. Otherwise, I’m not convinced and I might reduce my scores.**
> >
> > I hope this is not surprising. My intention is that the paper proves the point.

---

> > > ### Author Response · Authors · 2023-11-20
> > > **Response to Reviewer EJrs' comment**
> > >
> > > We thank the reviewer for taking the time to read and engage with us in our response. Below we seek to address the reviewer’s concern with regards to the complexity of the dataset. We analyze the complexity from several dimensions:
> > >
> > > 1. **Lexical diversity**: We use generally accepted measures [1] to analyze the diversity of datasets. We compute 1/2/3-gram entropy and the mean segmented token type ratio (MSTTR). To establish a comparison, we compute these values for three other datasets: XSUM (extreme summarization of news articles), DialogSum (real-life scenario dialogue summarization), and TinyStories. These have been specifically picked as they are stylistically different from our goal and script setup, and they often contain longer more natural sentences. We observe that even though the goals and steps in our dataset are shorter, the overall lexical diversity of Proscript and Coplan are comparable with other datasets. XSUM displays a higher MSTTR score, but this is likely attributed to the characteristics of news and more formal text. We also note that the machine-generated CoPlan exhibits slightly higher lexical diversity than the human-written proScript.
> > > 2. **Perplexity of an LM**: We also report the perplexity of an off-the-shelf language model which measures the degree of uncertainty (surprise) of an LM when it generates the next tokens. The higher the perplexity, the more surprised the LM is. As we see from the last column, with the exception of XSUM, the remaining datasets exhibit comparable perplexity scores. The higher perplexity in the case of XSUM is, again, attributed to its association with the news domain, a distinct characteristic compared to the other datasets, which predominantly encompass everyday scenarios. Also, intuitively LMs generally have lower perplexity scores on machine-generated data (as seen for CoPlan vs proScript).
> > >
> > >
> > > | Dataset     |   H1  |   H2  |   H3  | MSTTR | Perplexity (GPT2-med) |
> > > |-------------|:-----:|:-----:|:-----:|:-----:|:--------------------:|
> > > | XSUM        | 10.45 | 15.49 | 17.15 |  0.80 |         18.55        |
> > > | DialogSum   |  9.04 | 14.38 | 16.73 |  0.66 |         13.20        |
> > > | TinyStories |  8.65 | 14.25 | 17.80 |  0.61 |         7.53         |
> > > | ProScript   |  8.86 | 13.67 | 15.61 |  0.59 |         12.93        |
> > > | CoPlan      |  9.27 | 15.00 | 18.05 |  0.64 |         10.05        |
> > >
> > >
> > > [1] The GEM Benchmark: Natural Language Generation, its Evaluation and Metrics (Gehrmann et al., GEM 2021)
> > >
> > >
> > > ### Re overlapping of plans in train and test set (Triviality):
> > > **Overlapping of the plans**: As suggested by the reviewer, here we analyze the amount of overlap between the steps in the train and test set. Per the reviewer’s suggestion, we identify the direct noun object of a given goal (e.g., “Carry [a plate] to the kitchen”) and remove it from the goal (i.e., “Carry to the kitchen”) as well as all the steps in the corresponding plan (e.g., “Pick up the plate” –> “Pick up”. We then concatenate each goal with individual steps (i.e., “Carry to the kitchen. Pick up.”) and measure the maximum longest subsequence match of goal+step in the test set over all goal+step in the train set.
> > >
> > > In the proScript dataset, we find that only **4.3% of steps in the test set have exact overlap** with steps in the train set. If we relax the overlap to 90% and 80% (as opposed to an exact 100% overlap), this number increases to 5.5% and 10%, respectively. If the same is computed for plan overlap (as opposed to the step level; i.e., the direct object removed goal and plan—not individual steps—are concatenated), we observe **0% overlap**.
> > >
> > > In the Coplan dataset, computing overlap w.r.t full train set is still in progress (due to its large scale). However, on a randomly sampled 10K instances from the train set, the numbers are **1.8%, 2%, and 3.3%** for exact, 90% and 80% **step overlap**, respectively. And **0% for plan overlap**.
> > >
> > > This suggests that differences in the object taken by the verb do not necessarily mean minimal changes to the plan. Intuitively speaking, this is sensical: changes in the direct object, in the real world, should affect the way we resolve a goal. For example, what extra steps we take (e.g., stack plates, but not mugs),  the final goal location of the item (e.g., kitchen sink for plates, but the fridge for apples), or the manner of carrying (e.g., glasses vs. boxes) will affect the steps we take in a plan even if the goal is constant (i.e. “Carry X to the kitchen”). This is even more pronounced in CoPlan as it contains a broad set of everyday human goals (see Appendix A) which can lead to vastly distinct plans even when the event (verb) itself is the same. While learning is learning, the differences between the path taken to “Learn to play a violin” vs. “Learn to play Monopoly” vs. “Learn to speak Spanish fluently” are non-trivially different.

---

> ### Author Response · Authors · 2023-11-20
> **(continued) Response to Reviewer EJrs' comment**
>
> ### Re Can a classifier solve the problem?
> - The original proScript paper proposed a **prediction task** to be: given a goal and an unordered list of steps, predict a set of ordered steps. (Note that in this setting the model has access to the ground-truth list of steps which is NOT the case with the end-to-end plan **generation task**.)
> - They implemented a two-step approach baseline where they trained a **binary classifier** (RoBERTa-large) to predict the precedence between pairs of steps, followed by building an ordered list of steps (plan) by aggregating the predicted relations across all pairs of steps. Scores by the classifier are used as weights to create an adjacency matrix which is then automatically converted into ordered steps (plan).
> - This classification baseline achieved an **F1 score of 61.20%** which is far behind **human performance of 89.28%**. This suggests that the dataset is not trivial for a classification model even with full access to ground-truth steps. In the generation task (our focus) where the model **ONLY** has access to the goal, the task becomes even more challenging.
>
>
> We include all this discussion on the complexity of the datasets in the Appendix of our updated pdf.
>
>
> ### Re details of crowdsourcing proScript:
> First, given a scenario (e.g., bake a cake), each crowd-worker is required to describe five to seven core events (to balance the cognitive load and cost) that are essential for the scenario with the estimated time it takes to complete each event. In the second question, crowdworkers confirm the set of steps and they are asked to create a flowchart by connecting the steps possibly in partial order. When crowd-workers make a submission, a validation function is executed to check if the created flowchart is a valid dag and does not contain any shortcut edge. To have micro and microscopic scenarios, they iteratively picked events and used them as an additional source of finer-grained scenarios. For example, “turn on the oven” is a new fine-grained scenario derived from “bake a cake”.
> This results in 6,414 (train=3,252) valid scripts that include 311,502 pairs of events.
>
>
> ### Re training data used for finetuning single and multitask models:
> We trained all (single task) PlaSma models on their respective subsets of the CoPlan dataset. PlaSma variants (besides Plasma-Mul) in Table 1 are trained on the goal-based planning subset of CoPlan. Similarly, PlaSma models in the left plot of Figure 4 are trained on the constrained planning subset of Coplan, and so on. PlaSma-Mul, on the other hand, is jointly trained on all CoPlan subsets: goal-based planning, constrained planning, and counterfactual replanning. We will briefly explain this in the main text and move hyperparameters in the appendix.
>
>
> ### Clarification of task settings:
> We would like to clarify that in the paper we used “goal-based planning” and “original planning task” interchangeably. These two are contrasted with the new proposed task settings: “constrained planning” and “counterfactual replanning”.
>
>
>
> We thank the reviewer for their thoughtful comments and discussion.

---

> ### Author Response · Authors · 2023-11-23
> **Awaiting additional feedback (if any)**
>
> As the discussion period is nearing its conclusion, we would appreciate knowing if there are outstanding matters subsequent to our discussion.
>
> We hope our response sufficiently addresses your concern. Your feedback is invaluable, and we are committed to enhancing the quality of our work. We have updated our document accordingly and summarized the changes in the general response.
>
> Once again, we appreciate your time and consideration.
> Best, Authors

---

### Official Review · Reviewer_h13v · 2023-11-02

**Soundness:** 2 fair
**Presentation:** 3 good
**Contribution:** 2 fair
**Rating:** 6
**Confidence:** 4

**Summary:**

The paper introduces a framework designed to improve the procedural knowledge and planning capabilities of small language models. This is achieved through symbolic procedural knowledge distillation and a verifier-guided step-wise beam search algorithm. The authors have conducted experiments to compare student models of varying sizes with their teacher model, and have utilized human evaluations to assess the generated plans in terms of sequence, completeness, and overall quality. The findings indicate that smaller models can reach or even surpass the performance of larger models by employing the PLASMA framework.

**Strengths:**

- The paper is well-written and well-structured.
- Equipping small language models to come up with procedural knowledge at the same level as large language models is an important direction from an engineering perspective given the accessibility, carbon footprint, and cost of large language models.

**Weaknesses:**

- Although human evaluations were conducted, the executability conditions for the plans in the domains used in these experiments seem to be loose. It would be beneficial to evaluate the models in domains which have hard executability conditions (like the domains used in International Planning Competitions), where the correctness can be objectively determined, to more accurately gauge the language planning abilities of the proposed method.
- A comparison with GPT-4, in addition to GPT-3, could provide additional insights into the effectiveness of the method.
- The potential for increased bias due to the distillation from larger language models is mentioned in the limitations section but remains a concern.

**Questions:**

- If smaller language models can be effectively paired with human input or external verifiers for improved planning, why is distillation from a larger model necessary? This question is particularly relevant given that the domains discussed in the paper appear to be amenable to human verification.

---

> ### Author Response · Authors · 2023-11-17
> **Response to Reviewer h13v**
>
> Thank you for your helpful review and positive feedback on the importance of our research to the community and our well-structured paper. We are in process of updating our pdf, meanwhile please see our response below:
>
> ### Re W1: evaluate the models in domains which have hard executability conditions:
>
> - We indeed conducted an extrinsic evaluation in a domain with hard executability conditions, i.e., **VirtualHome (Section 3.3)**. This also helps to investigate the application of PlaSma and its generalization to the embodied agent planning domain. Given our strong results in the VirtualHome environment, we expect that PlaSma can serve as a strong foundation model and be easily transferred to other embodied domains with minimal domain adaptation.
> - On the general domain (i.e., on CoPlan); however, we follow the common practice of Likert scale human evaluation [1-3] across multiple dimensions. However, we made sure these dimensions are aligned with plan ability in an embodied domain. For example, **temporal ordering** is related to **executability** (e.g., `grasping` should be done before `picking up`), and **coverage/completeness** is related to success/correctness.
>
> We hope this clarifies the reviewer concern regarding our evaluation.
>
> [1] Recent Advances in Neural Text Generation: A Task-Agnostic Survey. (Tang et al., 2023)
>
> [2] Human evaluation of automatically generated text: Current trends and best practice guidelines (Van Der Lee et al., CSL 2021)
>
> [3] The use of rating and Likert scales in Natural Language Generation human evaluation tasks: A review and some recommendations (Amidei et al. iNLG 2019)
>
> ### W2: comparison with GPT-4:
> Our data was collected using GPT-3 Curie (smaller and less powerful than GPT-4) as the teacher due to cost constraints. We do expect to get better results for our distillation if using GPT-4 as our teacher. Nonetheless, as suggested by the reviewer, we ran a human evaluation comparing GPT-4, GPT-3.5, our best PlaSma model and its teacher on 50 instances (total of 200):
>
> |                         | Coverage | Order | Overall |
> |-------------------------|----------|-------|---------|
> | Fewshot Curie (teacher) | 3.53     | 4.37  | 3.58    |
> | PlaSma (best)           | 4.31     | 4.68  | 4.23    |
> | GPT-3.5 (Davinci-003)   | 4.78     | 4.84  | 4.81    |
> | GPT-4                   | 4.78     | 5.00     | 4.81    |
>
> As we observe, the trend remains the same, with GPT-4 surpassing its predecessor (GPT-3.5) only in the `ordering` dimension. We will add this comparison in the **Appenidx F** of our updated pdf.
>
> ### W3: Re potential increased bias:
> We acknowledge potential biases, and  have discussed  them in the limitations section, warranting the need for further investigation in  future research.
>
> ### Q1: If smaller language models can be effectively paired with human input or external verifiers...?
>
> Smaller models without distillation are not competent to perform the planning task and thus do not benefit from being augmented with an external verifier or human input. In fact, in our initial experiments, we tried to get an undistilled T5 model to work on the planning task. However, trying several prompt formats (such as “How to <goal>?”, Provide steps to achieve the <goal>”, etc.), we couldn’t get any meaningful outputs from this baseline to run human evaluation on or apply guided decoding. Most outputs were just repeating the given inputs or empty strings. However, we successfully ran a few-shot variant of undistilled T5 (with some post-processing to format the outputs). Below is the human evaluation result for this baseline on the same 250 randomly sampled test examples:
>
> |                         | Coverage | Order | Overall |
> |-------------------------|----------|-------|---------|
> | Fewshot T5 (no distillation) | 2.63     | 3.05  | 2.63    |
> | Fewshot Curie (teacher) | 3.75     | 4.27  | 3.75    |
> | PlaSma (best)           | 4.53     | 4.77  | 4.58    |
>
> Results of Few-shot undistilled T5 is significantly behind those of fewshot Curie (teacher) and all our model variants (**Table 1**).

---

> > ### Author Response · Authors · 2023-11-22
> > **Response to Reviewer h13v**
> >
> > As the discussion period is nearing its conclusion, we would appreciate knowing if there are any additional questions or outstanding matters subsequent to our discussion. We are happy to provide as much clarification as possible within this concluding phase.
> >
> > If you think our responses and updates have addressed your concerns, could you please consider raising your rating of the paper? Thank you!

---

> > ### Comment · Reviewer_h13v · 2023-11-23
> >
> > Thank you for taking your time and writing the response. Most of my concerns have been addressed and I have updated the score of the paper. My concern regarding the evaluation on domains where the correctness can be objectively determined as opposed to humans evaluating it (which might not be accurate to gauge the planning abilities) still remains.

---

### Official Review · Reviewer_drVQ · 2023-11-07

**Soundness:** 3 good
**Presentation:** 2 fair
**Contribution:** 3 good
**Rating:** 6
**Confidence:** 4

**Summary:**

This paper proposed a distillation procedure and an inference-time decoding algorithm to enable relative small language models for planning and replanning with performance close or surpassing its larger language teacher models.

**Strengths:**

• Proposed a paradigm to distill procedural planning knowledge from large language models to enable smaller languages to do planning, and it seems to be working.
	• Within the paradigm, a cost-effective human-in-loop LLM generated data curation procedure is also proposed to create the COPLAN dataset.
	• Proposed a  guided decoding procedure with a LLM-based (RoBERTa) step verifier to  guide the beam-search during planning steps decoding generation. The guidance help to further regulate the validity of the steps.

**Weaknesses:**

• The LLM-to-Planning-Model teacher-student paradigm for planning is not well motivated. Cost, performance (from specialization), controllable procedure, better-integration with downstream tasks (e.g. decoding/execution) and so on? It is more about better understanding of the key capabilities of existing techniques and combining them to solve the critical problems. For example, if it is more about specializing common knowledge embedded in LLMs to do planning, then smaller LLM might not be the right solution --- the same proposed paradigm can be combined with LLM of the same size or even larger LLMs for superior planning capabilities. What are the real problems and the corresponding means could be better sorted out?
	• The truth contribution and their relevancy might be hidden in the paper title and the current way of writing.  The proposal is composed of three parts (1) planning data generation from LLMs with human-in-loop curation, (2) teacher-student distillation training, (3)  language model decoding with step verifier.  There are less texts regarding teacher-student distillation. This might indicate that the teacher-distillation importance is over-estimated. With the planning data generation and verifier-guided decoding generation, there might other ways to enhance planning abilities, e.g. finetuning the original LLMs to specializing into planning domains. If the distillation step is an importance component in the ingredients, please detail it and discuss more.

**Questions:**

1. There are good ideas within the paper. The writing could be improved to make these good idea clear and stand-out. For example, how to train the step-verifier from human-written plans along with more formal analysis of impact of the step-verifier.
	2. Please define the loss functions formally with teacher-student distillation and verifier-training.
	3. For the step verifier, "we design perturbations … ordering, semantic completeness, topicality and fluency", please provide detailed analysis of these data-side steps regarding their intuition and formal properties if possible. How does a single verifier score reflect all these criteria? Any special design to achieve them with a simple RoBERTa based classifier?
	4. Regarding the evaluation metrics, please provide more details of the AMT human steps. Are coverage, order, over quality complete to evaluate a plan? Any comparison or correlation on the human evaluation metrics and the bleu numbers and the Emobided Environment's metrics? If not well-correlated, any proposal on automatic evaluating plans? Also how to relate and align human evaluation, bleu-style sequence matching metrics, embodied environment testable metrics and real-world execution measurable metrics?

---

> ### Author Response · Authors · 2023-11-17
> **Response to Reviewer drVQ**
>
> Thank you for the detailed review and positive comments on our cost-efficient distillation paradigm and our verifier-guided decoding algorithm. We are working on improving the writing of the paper to better showcase the ideas/contributions as suggested by the reviewer. Meanwhile, please find our responses below:
>
> ### Re importance of distillation:
>
> Finetuning LLMs requires updating models’ parameters which is not only costly but often inaccessible for the broader community. To clarify, our goal is not to make superior large models, but to distill procedural knowledge of LLMs into smaller models.
> Reducing the scale and cost of strong models via teacher-distillation is the key in developing open-sourced LMs that are accessible to all, facilitating fine-tuning and seamless adaptation to various domains and custom use cases. Given the large-scale training dataset used for PlaSma, we hope it can serve as a foundation model that can be quickly adapted to specific domains with minimal additional annotation (like we demonstrate by adapting Plasma to VirtualHome).
> Moreover, we could not augment most of the LLMs with our decoding-time algorithm due to limited access to the model's log probabilities.
>
> Thanks for the comment. We will update our pdf to include this.
>
> ### Q1. Re details of training the verifier and perturbation strategies:
> We included implementation details of the verifier in **Appendix B.3** due to space limit. We will move these materials to the main text as we recognize the potential inconvenience:
>
> - For training, we reuse 3k human-written plans from the existing ProScript dataset [2] to automatically create 47K positive and negative pairs of (plan-so-far, next-step).
> - Our perturbation strategies to generate pseudo-negative examples include: **(1) Reordered steps**: Conflicting logical order results from inaccurate causal or temporal dependencies in a plan. Thus, we apply both near and distant reordering by randomly reordering two consecutive and two distant steps. **(2) Repetitive steps**: Degeneration i.e., generating repetitive text is commonly observed in language models (also observed in the planning domain [1]). Similarly, we include both near and distant repetition by repeating the immediate previous step and distant previous step as a pseudo-negative next-step. **(3) Missing steps**: Another common mistake made by language models is missing necessary steps, leading to (semantically) incoherent plans. To simulate this behavior, we randomly select a non-immediate step as a pseudo-negative next-step. Below we show symbolic examples of perturbed pairs:
>
> Goal: g
>
> Oracle Plan (steps): s1 s2 s3 s4 s5
>
>
> We have the following perturbations:
>
> Reordering: (g + s1 + s3, s2) → 0
>
> Repetitive: (g + s1 + s2 + s3,  s2)  → 0
>
> Missing: (g + s1 + s2, s4) → 0
>
>
> Note that our perturbation strategies follow general rules applicable to any new domain with some task-specific adaptation.
>
> - Using the constructed positive and negative pairs of (plan-so-far, next-step), the verifier is trained using the standard binary Cross Entropy loss to identify the validity of the candidate next-step. It achieved an F1 score of 78% on a held-out test set.
>
> ### Re formal analysis of impact of the step-verifier:
> In Table 1, we include several ablations to show the contribution of multitasking, **step-verifier**, and scale. For example, the comparison between PlaSma and PlaSma+ across different model sizes shows non-negligible performance boosts as a result of the step verifier in our proposed tree-based decoding.
>
> Is the reviewer seeking a specific formal analysis? Thanks.
>
>
> ### Q2. Re defining loss functions:
> All models are trained with the standard Cross Entropy (CE) loss function with an early stop based on the validation loss. We will update our manuscript soon to include the loss used for distillation and verifier training.
>
> ### Q3. Verifier perurtabation strategies
> Please see our response to Q1

---

> ### Author Response · Authors · 2023-11-17
> **Response to  Reviewer drVQ (Continue)**
>
> ### Q4.1 Are human evaluation dimensions enough?
> Given the open-ended nature of CoPlan goals having varying levels of actionability (not all of them are grounded), we follow the common practice of Likert human evaluation across multiple dimensions [1-3]. Notably, we make sure we evaluate plans on dimensions that are related to actual plan ability in an embodied domain. For example, temporal ordering is related to executability (e.g., grasping should be done before picking up), and coverage/completeness is related to success/correctness. Nonetheless, we believe that extrinsic evaluation is crucial; for this reason, we use **VirtualHome** to investigate the application of PlaSma and its generalization to the embodied agent with hard executability **(Section 3.3)**. Given our strong VH results, we expect that PlaSma can serve as a strong foundation model and be easily transferred to other embodied domains with minimal domain adaptation.
>
> ### Q4.2. Correlation between human and BLEU scores:
> We indeed compute the correlation between BLEU metric and human scores. We find that BLEU has very weak correlations to human scores of coverage, ordering an overall quality, with a **Pearson correlation of 7.7%, 5.9%, and 5.6%**. This verifies the fact that n-gram-based metrics may not provide an informative measure of performance (as also suggested by previous works [3,4])
>
> ### Q4.3: Proposal for better automatic evaluation:
> We appreciate the interesting questions raised by the reviewer regarding human and automatic evaluation alignment. We also acknowledge the need for better automatic evaluation of plans, ideally those that take into account the causal, temporal, and completeness of steps towards achieving a goal. However, these are out of the scope of the current project and we hope future works can explore this direction.
>
> [1] Huang et al. “Language models as zeroshot planners: Language Models as Zero-Shot Planners: Extracting Actionable Knowledge for Embodied Agent.” ICML 2022.
>
> [2] Sakaguchi et al. “proScript: Partially ordered scripts generation”. In Findings of the Association for Computational Linguistics: EMNLP 2021.
>
> [3] Novikova et al. “Why We Need New Evaluation Metrics for NLG” EMNLP 2017
>
> [4] Celikyilmaz et al. “Evaluation of Text Generation: A Survey“

---

> > ### Author Response · Authors · 2023-11-22
> > **Response to reviewer drVQ**
> >
> > As the discussion period is nearing its conclusion, we would appreciate knowing if there are any additional questions or outstanding matters subsequent to our discussion. We are happy to provide as much clarification as possible within this concluding phase.
> >
> > If you think our responses and updates have addressed your concerns, could you please consider raising your rating of the paper? Thank you!

---

> > > ### Comment · Reviewer_drVQ · 2023-11-23
> > >
> > > Thank the authors for the efforts to answer other reviewers' and my concerns. Some of my concerns are addressed. My concern over (1) the rigidity of the evluation, and (2) the necessity of the distillation step in this paper remains. Related to the rigidity, the dataset issue reaised by other reviewer also concerns me. So I will keep my original rating. As I mentioned in my original review, there is good idea in this paper, please keep improving this work.

---

> ### Author Response · Authors · 2023-11-23
> **Response to Reviewer drVQ**
>
> We thank the reviewer for taking the time to read and engage with us in our response.
>
> Concerning the rigidity of the evaluation, we took substantial steps to address this issue. Firstly, we conducted an **extensive human evaluatio**n, scrutinizing various facets of a plan within a sizable subset of the test set (300 examples). Additionally, we undertook an **extrinsic evaluation on VirtualHome**, a platform characterized by **hard executability conditions**, to delve deeper into PlaSma's effectiveness within an embodied environment. We achieved **94.18% executability** rate compared to previous baseline of 77.17% and 43.68% success rate compared to 18.33%.  These show the adaptability of our model to new domains quickly.
>
> To provide more clarity and enhance the robustness of our work, we would appreciate specific feedback on the aspects of the evaluation that remain worrisome.
>
> ---
>
> Re necceessity of distillation: The essence of our approach lies in reducing the scale and cost of potent models through teacher-distillation. This, in turn, contributes to the development of open-sourced language models that are accessible to a wider audience, **enabling seamless fine-tuning and adaptation to diverse domains and custom use cases** (as we showed empirically). As we mentioned, fine-tuning LLMs requires updating parameters, a process both costly and often infeasible for broader community.
>
> ---
>
> Furthermore, regarding the rigidity of the dataset, an issue raised by another reviewer. We have addressed this concern by **incorporating several analyses/results that highlight the non-trivial nature** of the dataset under examination. (included in Appendix G of our updated document)
>
> ---
>
> As we continue to refine our work, we are committed to incorporating constructive feedback and further improving the merit of our contributions. We appreciate the recognition of the underlying ideas of our paper and remain dedicated to enhancing the quality of our research.
>
> Thanks again, authors

---

### Comment · Reviewer_EJrs · 2023-11-17
**Our comments**

Dear Authors, there is less than one week left for the discussion. I was more positive than other reviewers about your paper. Please participate in the discussion soon if you are willing to try to convince us further about the merit of the work.

---

> ### Author Response · Authors · 2023-11-17
>
> Thank you so much for your positive comments and detailed reviews. We were waiting for some (human) experiment results. We just submitted our responses.
>
> Please let us know if you have any further questions/concerns and we are more than happy to continue the discussion.
>
> Thanks, authors

---

### Author Response · Authors · 2023-11-20
**General response highlighting our contributions**

We value all the helpful comments/suggestions by the reviewers. We ensure their incorporation in our updated pdf.

As the discussion period ends soon (in 2 days), we'd like to summarize and highlight the contribution of our work as follows:

Our novelty lies in **(1)** the design of a unified and generalizable framework for data collection, distillation, and inference-time algorithm, **(2)** proposed novel counterfactual setting in the context of language-based agents, and **(3)** the application of PlaSma as a small foundation model in embodied environments. More specifically:

- We propose a unified framework for distilling procedural knowledge encoded into smaller planners with broader applications. As LLMs are gaining more attention in robotics and physically embodied applications, (1) this work can open up new **opportunities for the broader community** to explore small LMs for grounded/embodied planning **(shown empirically in Sec. 3.3)**, (2) our novel counterfactual tasks, datasets, and **enhancement of smaller LMs with counterfactual and contained (re)planning** capabilities offers a valuable and unique perspective.
- Our proposed **verifier-guided decoding** differs from existing decodings in that they mostly assume access to per-token logits/gradient updates and are applied at the token level. In contrast, our method only relies on model log-probabilities and a verifier and enforces semantic and temporal constraints at a **step level**.
- PlaSma achieves compelling results compared to much larger models.

---

### Author Response · Authors · 2023-11-22
**summary of updates to the paper:**

We thank the reviewers and area chair for carefully considering our work. Below we list major changes, clarifications, and improvements to our paper as part of the rebuttal, with changes colored red in the updated document:

- The addition of a brief description of perturbation strategies for verifier training in Sec 2.3 and refer to the detailed description in Appendix B.3. (Reviewer **drVQ**)
- The addition of the loss function used for training distilled models in Sec. 2.3 (Reviewer **drVQ**)
- Reporting the correlation between human scores and bleu scores in Appendix D.1 . (Reviewer **drVQ**)
- Conducting comparison with GPT-4 and including results in Appendix F (Reviewer **h13v**)
- Clarification of dataset used for experiments in Table 1 and Figure 4. (Reviewer **EJrs**)
- The addition of a thorough complexity and diversity analysis of proScript and CoPlan datasets in Appendix G  (Reviewer **EJrs**)
- An extended discussion on the importance of distillation, and the term “symbolic” in Appendix H (Reviewer **drVQ, EJrs, bpqq**)
- An extended related work in Appendix I discussing relevant work (Reviewer **bpqq**)

Thank you! authors

We believe that the vast majority of reviewer concerns have been addressed, either by changes to the paper draft or through discussion. We ensure the inclusion of other suggestions in our final draft.

---

### Meta-Review · Area_Chair_1wVs · 2023-12-24

**Metareview:**

### Summary
This paper proposed a distillation procedure and an inference-time decoding algorithm to enable relative small language models for planning and replanning with performance close or surpassing its larger language teacher models.
The model leverages symbolic procedural knowledge distillation and an inference-time algorithm, to enable smaller models with enhanced procedural knowledge and planning capabilities.

###  Strengths
+ Equipping small language models to come up with procedural knowledge at the same level as large language models
+ Reduces the scale and cost of potent models through teacher-distillation
+ Proposed a guided decoding procedure with a LLM-based step verifier

### Weaknesses:
- Lack of clarity and self-containments in writing,
- Limited evaluation in terms of domains for procedural knowledge.

**Justification For Why Not Higher Score:**

The paper draft needs improvements to clarify the reviewer questions and additional experiments.

**Justification For Why Not Lower Score:**

By leveraging symbolic procedural knowledge distillation and an inference-time algorithm, we have endowed smaller models with enhanced procedural knowledge and planning capabilities.

---

### Decision · Program_Chairs · 2024-01-16

Accept (poster)